# Information Directed Reward Learning for Reinforcement Learning

**David Lindner**
Department of Computer Science
ETH Zurich
david.lindner@inf.ethz.ch

**Matteo Turchetta**
Department of Computer Science
ETH Zurich
matteo.turchetta@inf.ethz.ch

**Sebastian Tschiatschek**
Department of Computer Science
University of Vienna
sebastian.tschiatschek@univie.ac.at

**Kamil Ciosek**[*]
Spotify
kamilc@spotify.com

**Andreas Krause**
Department of Computer Science
ETH Zurich
krausea@ethz.ch

## Abstract

For many reinforcement learning (RL) applications, specifying a reward is difficult. This paper considers an RL setting where the agent obtains information about the reward only by querying an expert that can, for example, evaluate individual states or provide binary preferences over trajectories. From such expensive feedback, we aim to learn a model of the reward that allows standard RL algorithms to achieve high expected returns *with as few expert queries as possible*. To this end, we propose *Information Directed Reward Learning* (IDRL), which uses a Bayesian model of the reward and selects queries that maximize the information gain about the difference in return between plausibly optimal policies. In contrast to prior active reward learning methods designed for specific types of queries, IDRL naturally accommodates different query types. Moreover, it achieves similar or better performance with significantly fewer queries by shifting the focus from reducing the reward approximation error to improving the policy induced by the reward model. We support our findings with extensive evaluations in multiple environments and with different query types.

## 1 Introduction

*Reinforcement learning* (RL; Sutton and Barto, 2018) casts the problem of learning to perform complex tasks by interacting with an environment as an optimization problem where the learning agent aims to maximize its expected cumulative reward. Despite the remarkable successes of RL (e.g., Mnih et al., 2015; Silver et al., 2016), specifying reward functions that capture complex tasks is still an open problem. A promising approach is to *learn* a reward function from human feedback (e.g., Christiano et al., 2017). However, since human feedback is expensive, *active reward learning* aims to minimize the number of queries. Prior work often focuses on approximating the reward function uniformly well. However, this may not be aligned with the original goal of RL: *finding an optimal policy*, as Figure 1 shows. Moreover, prior work is often tailored to

---

[*]Work done while at Microsoft Research Cambridge.

35th Conference on Neural Information Processing Systems (NeurIPS 2021).

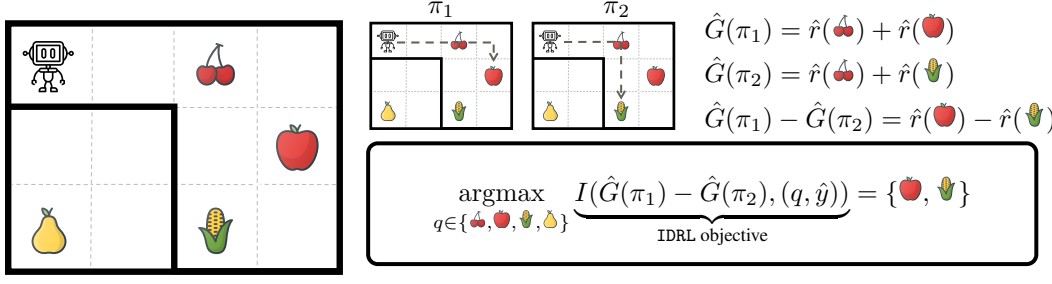

$T = 4$

Figure 1: The robot wants to collect food for a human. It can only move 4 timesteps in the gridworld, cannot pass through the black walls, and collecting more food is always better. The robot does not know the human's preferences, but it can ask for food ratings. Common active learning methods aim to learn the reward uniformly well, and would query all items similarly often. In contrast, IDRL considers only the two plausibly optimal policies $\pi_1$ and $\pi_2$. Since both policies collect the cherry, and do not collect the pear, the robot only needs to learn about the apple and the corn. IDRL can solve the task with 2 queries instead of 4.

specific types of queries, such as comparisons of two trajectories (e.g., Sadigh et al., 2017) or numerical evaluations of trajectories (e.g., Daniel et al., 2015), limiting its applicability.

**Contributions.**   We propose *Information Directed Reward Learning* (IDRL), a general *active reward learning* approach for learning a model of the reward function from expensive feedback with the goal of *finding a good policy* rather than uniformly reducing the model's error. IDRL can use arbitrary Bayesian reward models and arbitrary types of queries (Section 4), making it more general than existing methods. We describe an exact and efficient implementation of IDRL using Gaussian process (GP) reward models (Section 5) and different types of queries, and an approximation of IDRL that uses a deep neural network reward model and a state-of-the-art policy gradient algorithm to learn from comparison queries (Section 6). We evaluate IDRL extensively in simulated environments (Section 7), including a driving task and high-dimensional continuous control tasks in the MuJoCo simulator, and show that both implementations significantly outperform prior methods.

## 2   Related work

**Reward Learning for RL.**   Several works aim to directly learn policies rather than reward functions from expert feedback in the form of numerical evaluations (Knox and Stone, 2009; MacGlashan et al., 2017) or comparisons (Regan and Boutilier, 2009; Fürnkranz et al., 2012), and some work also explores active query selection (Akrour et al., 2012; Wilson et al., 2012). However, learning policies directly from feedback has several downsides: it is difficult to combine different types of feedback, and policies tend to generalize poorly between environments. Ng and Russell (2000) argue that reward functions are a more robust representation of desired behavior than policies. *Inverse reinforcement learning* (IRL) aims to learn a reward model from expert demonstrations (Abbeel and Ng, 2004). Reward models can also be learned from comparisons of two or more different behaviors (Wirth et al., 2017), or other kinds of feedback (Jeon et al., 2020). While reward models can successfully learn hard-to-specify control and game-playing tasks (Christiano et al., 2017; Ibarz et al., 2018), most work uses simple heuristics to select queries to make. In contrast, *active reward learning* aims to select the most informative queries in a principled way.

**Active reward learning.**   For linear reward functions, Sadigh et al. (2017) ask the expert to compare trajectories synthesized to maximize the volume removed from a hypothesis space. Bıyık et al. (2020b) argue that maximizing information gain leads to better sample efficiency and queries that are easier to answer than volume removal. Bıyık et al. (2020a) generalize maximizing information gain to non-linear reward functions using a GP model. We also use an information gain objective to select queries; however, our approach focuses on finding an optimal policy instead of uniformly reducing the error of the reward model. With a similar motivation, Wilde et al. (2020) aim to capture how informative a query is for distinguishing policies. However, their method is limited to comparisons between potentially optimal policies. Daniel et al. (2015) also introduce an acquisition function to measure how informative a query is for learning a good policy. However, their setting is restricted to observing the cumulative reward of a trajectory, and their acquisition function is computationally expensive. Table 1 gives an overview of how our method compares to this prior work.

| | Non-linear Rewards | Single-state Queries | Trajectory Queries | Numerical Queries | Comparison Queries | Considers Env. Dynamics |
|---|---|---|---|---|---|---|
| Sadigh et al. (2017) | ✗ | ✗ | ✓ | ✗ | ✓ | ✗ |
| Bıyık et al. (2020b) | ✗ | ✗[1] | ✓ | ✗[1] | ✓ | ✗ |
| Bıyık et al. (2020a) | ✓ | ✗[1] | ✓ | ✗[1] | ✓ | ✗ |
| Daniel et al. (2015) | ✓ | ✗[1] | ✓ | ✓ | ✗[1] | ✓ |
| Wilde et al. (2020) | ✗ | ✗ | ✓ | ✗ | ✓ | ✓ |
| IDRL (ours) | ✓ | ✓ | ✓ | ✓ | ✓ | ✓ |

[1] The original authors do not consider this setting, but we provide an extension to their method in Section 7 and Appendix D.

Table 1: In contrast to most prior work on active reward learning for RL, IDRL can handle non-linear reward functions and different query types, in particular numerical evaluations and comparisons of individual states and (partial) trajectories. Further, IDRL takes the environment dynamics into account to achieve better sample efficiency (cf. Figure 1).

**Bayesian optimization.** *Bayesian optimization* (BO) aims to maximize an expensive-to-evaluate function by learning a Bayesian model of the function and selecting informative queries (Mockus et al., 1978). We face a related but significantly harder problem: we aim to find an optimal policy in RL but only *indirectly* obtain information about the value of policies. Our problem has striking connections to variants of the *multi-armed bandit problem* (Bubeck and Cesa-Bianchi, 2012), in particular *partial monitoring problems* (Rustichini, 1999) and *transductive linear bandits* (Fiez et al., 2019). We explore this connection in detail in Appendix C.

## 3 Background and problem setting

**Markov decision process.** Markov decision processes (MDPs; Puterman, 2014) model sequential decision-making problems in dynamical systems. An MDP $(\mathcal{S}, \mathcal{A}, P, r, p_0, \gamma)$ consists of a state space $\mathcal{S}$, an action space $\mathcal{A}$, a transition function $P$, a reward function $r$, an initial state distribution $p_0$, and a discount factor $\gamma \in [0, 1)$. In an MDP, the agent starts in state $s_0 \sim p_0(s)$ and, when taking action $a_t$, transitions from state $s_t$ to state $s_{t+1}$ with probability $P(s_{t+1}|s_t, a_t)$. The agent affects the environment through actions determined by a policy $\pi(a_t|s_t)$, indicating the probability of taking action $a_t$ in state $s_t$. The agent's goal is to find a policy $\pi$ that maximizes the *expected discounted return* $G(\pi) = \mathbb{E}_{P,\pi,p_0} [\sum_{t=0}^{\infty} \gamma^t r_t]$, where $r_t$ is the reward obtained at time $t$.

**Information gain.** Intuitively, the information gain between two random variables measures the amount of information that can be obtained about one of them by observing the other. Formally, for two random variables $X$ and $Y$ with marginal distributions $p_X$, $p_Y$ and joint distribution $p_{(X,Y)}$, the *information gain* (or mutual information) is $I(X,Y) = D_{\text{KL}}(p_{(X,Y)}\|p_X \cdot p_Y)$, where $D_{\text{KL}}(\cdot\|\cdot)$ is the KL-divergence. Given a third random variable $Z$, conditional information gain is defined as $I(X; Y|Z = z) = D_{\text{KL}}(p_{(X,Y)|Z=z}\|p_{X|Z=z} \cdot p_{Y|Z=z})$.

**Problem setting.** We focus on MDPs where the reward function is not readily available. Instead, the agent can query an expert for information about the reward. In iteration $i$, the agent makes a query $q_i$ to the expert, and receives a response $y_i$. For example, $q_i$ could ask the expert to compare two trajectories or judge a single trajectory, and $y_i$ could indicate which of the two trajectories is better or provide the return of a single trajectory. We assume that the agent can interact with the environment cheaply, but queries to the expert are *expensive*, and hence the agent has to find a policy $\pi$ that *maximizes the expected return* $G(\pi)$ using *as few queries as possible*.

**Our reward learning approach.** We approach this problem by learning a model of the reward function, i.e., a model that predicts the reward of a given state,[2] and computing a policy that maximizes the return induced by the model. Importantly, we want to learn a reward model such that the induced optimal policy achieves a high return under the true reward function. Note that any RL algorithm can be used to find the policy. Hence, the problem reduces to selecting a model for the reward function and deciding which queries to make. Our key insight is that queries that help most to find a good policy might differ from those that uniformly reduce the model's uncertainty.

---

[2]Our approach is also applicable to reward functions that depend on state-action pairs or transitions. We focus on state-dependent reward functions for simplicity of exposition.

**Algorithm 1** *Information Directed Reward Learning* (IDRL). The algorithm requires a set of candidate queries $\mathcal{Q}_c$, a Bayesian model of the reward function, and an RL algorithm that returns a policy given a reward function. $\hat{G}(\pi)$ is the belief about the expected return of policy $\pi$, induced by the reward model $P(\hat{r}|\mathcal{D})$, and $\hat{r}$ is the belief about the reward function.

---

1: $\mathcal{D} \leftarrow \{\}$; $\Pi_c \leftarrow$ initialize candidate policies; initialize reward model with prior distribution $P(\hat{r})$
2: **while** not converged **do**
3:      Select a query:
4:          $\pi_1, \pi_2 \in \mathrm{argmax}_{\pi,\pi' \in \Pi_c} H(\hat{G}(\pi) - \hat{G}(\pi')|\mathcal{D})$
5:          $q^* \in \mathrm{argmax}_{q \in \mathcal{Q}_c} I(\hat{G}(\pi_1) - \hat{G}(\pi_2); (q, \hat{y})|\mathcal{D})$
6:      Make query and update reward model:
7:          $y^* \leftarrow$ Response to query $q^*$
8:          $P(\hat{r}|\mathcal{D} \cup \{(q^*, y^*)\}) \propto P(y^*|\hat{r}, \mathcal{D}, q^*)P(\hat{r}|\mathcal{D})$        ▷ Update belief about the reward
9:          $\mathcal{D} \leftarrow \mathcal{D} \cup \{(q^*, y^*)\}$        ▷ Add observation to dataset
10:      Optionally update candidate policies $\Pi_c$
11: **end while**
12: $\bar{r} \leftarrow$ mean estimate of the reward model; $\bar{\pi}^* \leftarrow \mathrm{RL}(\bar{r})$
13: **return** $\bar{\pi}^*$

---

## 4   The Information Directed Reward Learning acquisition function

This section introduces Information Directed Reward Learning (IDRL) for a general Bayesian model of the reward and discusses how to select queries $q_i$, making no assumptions on their form nor on the responses $y_i$.

**Reward model.** To select informative queries, we need to quantify uncertainty; hence, we use a Bayesian model of the reward function. From a Bayesian perspective, it is important to distinguish between the agent's belief about a quantity and its "actual" value that is unknown to us. We denote the belief about the reward in state $s$ with $\hat{r}(s)$ and its actual value with $r(s)$.

**Query selection.** To select informative queries, we have to consider that the responses might only give *indirect* information about the set of optimal policies. For example, assume the agent can ask the expert to quantify the reward of individual states. These rewards provide information about the expected return of a policy but may yield no information about the set of optimal policies. For example, if a state is visited similarly often by every plausibly optimal policy, knowing its reward does not help decide between the policies (e.g., the cherry in Figure 1). Therefore, any approach that only aims to reduce the uncertainty of the reward model may waste expensive queries that do not help find an optimal policy.

Intuitively, we want to instead select queries that *help identify the optimal policy*. In the language of information theory, we want to maximize the *information gain* of a query about *the identity of the optimal policy*. More formally, if $\mathcal{D} = \{(q_1, y_1), \ldots, (q_t, y_t)\}$ is a dataset of past queries and responses, let us denote with $P(\hat{\pi}^*|\mathcal{D})$ the agent's belief about the optimal policy, induced by our belief about the reward function $\hat{r}$. Also, let $\mathcal{Q}_c$ be a set of candidate queries the agent can make. Then, one way to formalize this intuition is to select queries $q^* \in \mathrm{argmax}_{q \in \mathcal{Q}_c} I(\hat{\pi}^*; (q, \hat{y})|\mathcal{D})$, where $\hat{y}$ is the agent's belief about the response it will get to query $q$, and $I$ denotes the information gain. Unfortunately, this objective has two undesirable properties. First, the agent has to keep track of a distribution over all possible policies to compute it, which is intractable in general. Second, reducing uncertainty about the optimal policy only matters as long as there are significant differences in the return of plausibly optimal policies. For example, if the agent identifies a set of plausibly optimal policies with similar returns, we care less about identifying exactly which policy is optimal, compared to when such policies have very different returns.

To address the first challenge, we obtain a finite set of *candidate policies* $\Pi_c$ that are *plausibly optimal* according to our Bayesian reward model. To address the second challenge, we select the most informative query for distinguishing policies in terms of their value.

Let us first discuss how to select queries, assuming a set of plausibly optimal policies $\Pi_c$ to be available. We can exploit the fact that the belief about the reward function $\hat{r}$ induces a belief about the expected return of policy $\pi \in \Pi_c$, denoted as $\hat{G}(\pi)$. Concretely, the expected return

of a policy can be computed as the scalar product $G(\pi) = \langle \mathbf{f}^\pi, \mathbf{r} \rangle$, where $\mathbf{f}^\pi$ is a vector of the (discounted) expected state-visitation frequencies of policy $\pi$ and $\mathbf{r}$ is a vector of rewards of the corresponding states. We can estimate $\hat{\mathbf{f}}^\pi$ from trajectories sampled using policy $\pi$, and then determine $\hat{G}(\pi)$ from $\hat{r}$.

Given $\hat{G}(\pi)$, `IDRL` proceeds in two steps. It first selects two policies that maximize the model's uncertainty about the difference in their expected returns:

$$\pi_1, \pi_2 \in \underset{\pi, \pi' \in \Pi_c}{\operatorname{argmax}} H(\hat{G}(\pi) - \hat{G}(\pi')|\mathcal{D}), \tag{1}$$

where $H$ is the entropy of the belief conditioned on past queries. To gather information about the distinction between $\pi_1$ and $\pi_2$, `IDRL` then selects queries that maximize the information gain about the difference in expected return between $\pi_1$ and $\pi_2$:

$$q^* \in \underset{q \in \mathcal{Q}_c}{\operatorname{argmax}} I(\hat{G}(\pi_1) - \hat{G}(\pi_2); (q, \hat{y})|\mathcal{D}). \tag{2}$$

Note that this is not the same as jointly maximizing the information gain about $\hat{G}(\pi_1)$ and $\hat{G}(\pi_2)$. Equation (2) prefers queries that help to distinguish $\pi_1$ and $\pi_2$ over queries that help to determine the exact value of $\hat{G}(\pi_1)$ and $\hat{G}(\pi_2)$. Assuming a set of optimal policies is contained in $\Pi_c$, reducing the uncertainty about the difference in returns within $\Pi_c$ will help to identify an optimal policy quickly. In particular, if there is no remaining uncertainty about the differences in return, we can clearly identify an optimal policy.

Let us now discuss how to obtain a set of candidate policies $\Pi_c$. For `IDRL` to select informative queries, $\Pi_c$ has to reflect the agent's current belief about optimal policies. `IDRL` uses Thompson sampling (TS, Thompson, 1933) as a flexible way to create $\Pi_c$. We implement TS by repeatedly sampling a reward function from the posterior reward model and finding an approximately optimal policy for this sampled reward function. This approximates sampling from the posterior distribution over optimal policies. Since TS is demanding, in our experiments, we investigate two effective alternatives to alleviate its computational burden: (1) we update $\Pi_c$ in regular intervals, rather than at every step, and (2) we start from the candidates computed in previous steps rather than starting the policy optimization from scratch.

Algorithm 1 shows the full `IDRL` algorithm. In each iteration, `IDRL` identifies two plausibly optimal policies with high uncertainty about their difference in return and then aims to reduce this uncertainty. We can stop the algorithm after a fixed number of queries, or by checking a convergence criterion, and return a policy $\bar{\pi}^*$ that is optimized for the current reward model.

Note that `IDRL` is agnostic to how the candidate queries $\mathcal{Q}_c$ are generated. Different applications might require different approaches to generating $\mathcal{Q}_c$. In our experiments, for example, we consider: using all possible queries in small environments, choosing states or trajectories to query from rollouts of the currently optimal policy $\bar{\pi}^*$, selecting queries from rollouts of the candidate policies, and selecting queries from trajectories of a pre-defined explorations policy. Importantly, all of these, and others, are compatible with `IDRL`.

## 5 An exact and efficient implementation of `IDRL` for GP reward models

Here, we describe one concrete implementation of `IDRL` using a Gaussian process (GP, Rasmussen and Williams, 2006) reward model and linear query types. These choices allow us to compute equations (1) and (2) exactly and efficiently.

**Reward model.** We model the reward function as a GP with (w.l.o.g.) a zero-mean prior distribution $\hat{r}(s) \sim \mathcal{GP}(0, k(s, s'))$ using a kernel $k$ which measures the similarity of states.

**Query selection.** We first show how to compute equations (1) and (2) if the posterior belief about the reward function is Gaussian. Then, we discuss a family of practically relevant query types that satisfy this assumption. We provide proofs for all results in Appendix A.

**Proposition 5.1.** If $\hat{r}(s)|\mathcal{D}$ is a GP, then $P(\hat{G}(\pi) - \hat{G}(\pi')|\mathcal{D})$ is Gaussian and:

$$\underset{\pi,\pi' \in \Pi_c}{\operatorname{argmax}} H(\hat{G}(\pi) - \hat{G}(\pi')|\mathcal{D}) = \underset{\pi,\pi' \in \Pi_c}{\operatorname{argmax}} \operatorname{Var}[\hat{G}(\pi) - \hat{G}(\pi')|\mathcal{D}]$$

$$\underset{q \in \mathcal{Q}_c}{\operatorname{argmax}} I(\hat{G}(\pi_1) - \hat{G}(\pi_2); (q, \hat{y})|\mathcal{D}) = \underset{q \in \mathcal{Q}_c}{\operatorname{argmin}} \operatorname{Var}[\hat{G}(\pi_1) - \hat{G}(\pi_2)|\mathcal{D} \cup \{(q, \hat{y})\}]$$

We can compute both variances analytically, enabling exact implementation of equations (1) and (2).

**Query types.**    To apply this result, we need $\hat{r}(s)|\mathcal{D}$ to be a GP, which is not the case for general observations $(q_i, y_i)$. If the queries are individual states, i.e., $q_i = s_i$ and $y_i = r(s_i)$, the problem is standard GP regression, and $\hat{r}(s)|\mathcal{D}$ is a GP (Rasmussen and Williams, 2006). More generally, a similar statement holds if the observations are *linear* combinations of rewards.

**Definition 5.1.** We call $q = (S, C)$ a *linear reward query*, if it consists of states $S = \{s_1, \dots, s_N\}$ and linear weights $C = \{c_1, \dots, c_N\}$, and the response to query $q$ is a linear combination of rewards $y = \sum_{j=1}^{N} c_j r(s_j) + \varepsilon$, with Gaussian noise $\varepsilon \sim \mathcal{N}(0, \sigma_n^2)$.

**Proposition 5.2.**    Let $q$ be a linear reward query. If the prior belief about the reward $\hat{r}(s)$ is a GP, then the posterior belief about the reward $\hat{r}(s)|(q, y)$ is also a GP.

Linear reward queries result in a particularly efficient implementation of IDRL. Of course, IDRL with a GP model could be extended to non-linear observations using approximate inference. However, it turns out that many commonly used query types can be modeled as linear reward queries, including the return of trajectories or comparisons of trajectories (see Appendix B).

# 6    A scalable Deep RL approximation of IDRL

GP models provide a convenient way to implement IDRL exactly. But, can IDRL also be used if we can not model the reward function as a GP? Moreover, can we scale it to large environments on the scale of typical Deep RL applications?

To address these questions, we propose a second implementation of IDRL using a deep neural network (DNN) reward model. To scale IDRL to large Deep RL scenarios, we integrate it into a policy optimization algorithm, similar to Christiano et al. (2017).[3]  In our experiments, we focus on comparison queries, but it is straightforward to extend the algorithm to other query types.

**Reward model.**    To model the reward function, we use adaptive basis function regression with DNNs, similar to Snoek et al. (2015). Concretely, we train a DNN from comparisons of short clips of the agents behavior using the Bradley-Terry model and $\ell_2$-regularization. We then treat the learned representation as a basis function and the final layer of the DNN as a maximum *a posteriori* (MAP) estimate of the parameters of a Bayesian logistic regression model. Finally, we approximate the full posterior using a Laplace approximation.

**Query selection.**    Because of the Laplace approximation, the posterior distribution of $\hat{r}(s)|\mathcal{D}$ is Gaussian, and we can compute equations (1) and (2) the same way we did for a GP reward model.

**Query types.**    Similar to Christiano et al. (2017), we consider queries $q_i = (\sigma_i^1, \sigma_i^2)$ that compare two segments of trajectories $\sigma_i^1$ and $\sigma_i^2$, where the user responds with their preference $y_i \in \{-1, 1\}$.

**Candidate policies.**    In large environments, it is infeasible to train new policies from scratch during the Thompson sampling step. To avoid this, we maintain a fixed set of policies that we update regularly, instead of training new policies from scratch whenever we receive new samples.

**Candidate queries.**    We generate candidate queries by rolling out the current policy optimized for the mean estimate of the reward model, as well as the candidate policies and uniformly sampling pairs of segments from the resulting trajectories.

**Full algorithm.**    We use a policy gradient algorithm to train a policy for the current reward model, and the candidate policies. Similar to Christiano et al. (2017), the agent queries comparisons following a fixed schedule in which the number of samples is proportional to $\frac{1}{T}$, where $T$ is the number of policy training steps, i.e., we provide more samples early during training and less later on. For more details, including full pseudocode for the Deep RL algorithm, see Appendix E.3.

---

[3]We provide a detailed comparison between our setup and Christiano et al. (2017) in Appendix E.3.

# 7 Experiments

We empirically test IDRL in several environments, ranging from gridworlds to complex continuous control tasks, and for several different query types, including numerical evaluations and comparisons of trajectories. Our evaluation covers most scenarios existing in the literature and shows that IDRL attains comparable or superior performance to methods designed for specific scenarios.

In all experiments, the agent's queries are answered with simulated feedback based on an underlying true reward function unknown to the agent. We usually evaluate the *regret* of a policy $\pi$ trained using the reward model, i.e., $G(\pi^*) - G(\pi)$ for an optimal policy $\pi^*$. If we do not know $\pi^*$, we approximate it with a policy trained on the true reward function.

We first validate that GP-based IDRL improves sample efficiency in simple gridworld environments for numerical and comparison queries (Section 7.2). Next, we consider the most common setup in the literature, that is, learning from comparisons of trajectories, and compare GP-based IDRL against alternative approaches in a driving simulator, proposed in prior work (Section 7.3). Then, we study another natural feedback type: ratings of clips of the agent's behavior. In this setting, we demonstrate how GP-based IDRL can be scaled up to bigger environments in the MuJoCo simulator (Section 7.4). Finally, we further demonstrate scalability by considering the Deep RL implementation of IDRL to learn standard MuJoCo tasks from comparisons of clips of trajectories, similar to Christiano et al. (2017) (Section 7.5).

For each environment, we choose our setup to be close to prior work to promote a fair comparison. This leads to some design choices, such as the RL solver or the query types, to differ between environments. As a side effect, this highlights IDRL's generality. Appendices D and E describe the experimental setup in more detail, and we provide code to reproduce all experiments.[4]

## 7.1 Baselines

We consider five baselines: (*i*) *Uniform sampling* selects queries from $\mathcal{Q}_c$ with equal probability. (*ii*) *Information gain on the reward* (IGR) selects queries that maximize information gain about the reward $I((q, \hat{y}); \hat{r}|\mathcal{D})$. For a GP model, this is equivalent to maximizing $\mathrm{Var}[\hat{y}|\mathcal{D}, q]$. Bıyık et al. (2020b) use IGR to learn rewards from comparisons of trajectories; however, it can be extended to other query types. (*iii*) *Expected improvement on the reward* (EIR) maximizes the improvement in the value of a query compared to the best observation so far, in *expectation*, and is a common acquisition function in BO (Mockus et al., 1978). EIR can not be applied to comparison queries. (*iv*) *Expected policy divergence* (EPD) is an active reward learning method introduced by Daniel et al. (2015), which makes queries that maximally change the current policy. Since EPD updates the policy for each potential observation, it is prohibitively expensive for large $\mathcal{Q}_c$. While EPD was introduced to query the return of trajectories, we extend it to other query types (cf. Appendix D). (*v*) *Maximum regret* (MR) is an acquisition function proposed by Wilde et al. (2020). It assumes access to a set of candidate reward functions and corresponding optimal policies. MR compares policies that perform well according to one reward function but poorly according to a different one. It can only be used with comparisons of full trajectories. We also tested *expected volume removal* (EVR, Sadigh et al., 2017) for comparison queries; however, we found it to get stuck often, which confirms the findings of Bıyık et al. (2020b). Note that IGR and EIR reduce uncertainty uniformly over the state space, while EPD and MR consider the environment dynamics.

## 7.2 Can IDRL improve sample efficiency by considering the environment dynamics?

We first validate our hypothesis that IDRL improves sample efficiency in small toy environments. Here, we highlight experiments in a set of *Gridworlds* similar to Figure 1. Appendix F presents two additional toy environments that isolate specific reasons why IDRL outperforms the baselines.

**Setup.** We consider $10 \times 10$ *Gridworlds* with randomly placed walls and objects with different rewards. The agent has to find the object with the largest reward. We consider queries about the reward of individual states, i.e., $q_i = s_i \in \mathcal{S}$ and $y_i = r(s_i)$, and comparison queries with $q_i = (s_{i1}, s_{i2})$ and $y_i \in \{-1, 1\}$. The candidate queries $\mathcal{Q}_c$ either consist of all states or all

---

[4]https://github.com/david-lindner/idrl

pairs of states. We use GP-based IDRL, with a kernel that encodes which objects are the same, and which are different. All experiments run for less than 1 hour on a single CPU.

**Results.** Figure 2 shows the regret of a policy trained on the reward model after different numbers of queries. IDRL finds better policies than the baselines with a limited number of queries because it focuses on regions of the state space relevant for finding the optimal policy. As shown in Figure 1, this improves sample efficiency over methods that uniformly reduce uncertainty, such as IGR and EIR. IDRL also outperforms EPD, because EPD's goal of selecting queries that maximally change the current policy is also misaligned with the goal of finding an optimal policy. We investigate EPD's specific failure modes in Appendix F.

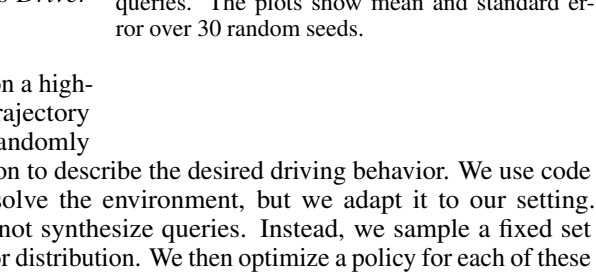

Figure 2: Regret of a policy trained in randomly generated gridworlds as a function of the number of queries. The queries are either asking about the reward of individual states, or comparisons between two states. The plot compares IDRL (●—●) to EPD (✕—✕), IGR (▲—▲) and uniform sampling (■—■). For numerical queries, we also compare to EIR (◆—◆), which is not applicable to comparisons. MR is not applicable to single-state queries. The plots show mean and standard error over 30 random seeds.

### 7.3 Can IDRL learn from comparisons of trajectories using a GP reward model?

Most prior work studies reward learning from comparisons of trajectories. To evaluate IDRL in this setting, we consider the 2-dimensional, continuous *Driver* environment by Sadigh et al. (2017).

**Setup.** In *Driver*, the agent controls a car on a highway with another car driving on a fixed trajectory (cf. Figure 3a). For each experiment, we randomly sample an underlying (linear) reward function to describe the desired driving behavior. We use code by Sadigh et al. (2017) to simulate and solve the environment, but we adapt it to our setting. In contrast to Sadigh et al. (2017), we do not synthesize queries. Instead, we sample a fixed set of 200 reward functions from a Gaussian prior distribution. We then optimize a policy for each of these reward functions, and, similarly to Wilde et al. (2020), consider all pairs of policies as potential queries. Moreover, we assume a linear observation model (see Appendix B), whereas Sadigh et al. (2017) and Wilde et al. (2020) choose different non-linear observation models. Each experiment runs for less than 24 hours on a single CPU.

**Results.** Figure 3a shows the regret curves for the learned policy and the cosine similarity for the learned reward function weights. IDRL outperforms the baselines and finds a better policy with fewer queries. However, the difference to pure information gain is small in this simple environment.

### 7.4 Can IDRL with a GP model be scaled to bigger environments?

To demonstrate that GP-based IDRL scales to larger environments, we use the MuJoCo simulator (Todorov et al., 2012), which provides challenging environments commonly used as benchmarks for RL. However, its standard locomotion tasks are very easy to learn for a GP model because the reward is directly proportional to the agent's velocity in x-direction. Instead, we propose a task where the reward function is harder to learn.

|  | *Swimmer-Corridor* | *Ant-Corridor* |
|---|---|---|
| Uniform Sampling | $11.8 \pm 0.9$ | $15 \pm 1$ |
| IGR | $13.3 \pm 0.6$ | $17.2 \pm 0.5$ |
| EIR | $12.0 \pm 0.8$ | $17.5 \pm 0.8$ |
| IDRL (20 updates) | $\mathbf{2.4 \pm 0.8}$ | $\mathbf{2.2 \pm 0.8}$ |
| IDRL (4 updates) | $2.8 \pm 0.7$ | $5 \pm 1$ |
| IDRL (2 updates) | $5.1 \pm 0.6$ | $8 \pm 1$ |
| IDRL (1 update) | $7.6 \pm 0.8$ | $12 \pm 1$ |

Table 2: Results comparing IDRL for different update frequencies of the candidate policies in the *Corridor* environments. The table shows the estimated regret of a policy trained using 20 queries about the reward function.

**Setup.** In our *Corridor* environments (Figure 3b), a robot (*Swimmer*, or *Ant*) has to move forward and stop at a goal position. The simulated expert rates trajectory clips according to a reward function that is proportional to the velocity in the direction of the goal. This reward function is linear in a set of features of the state, as described in Appendix D.3.5.

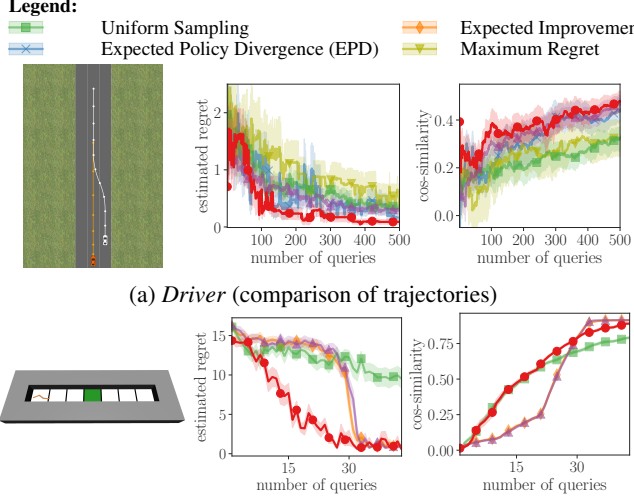

(a) *Driver* (comparison of trajectories)

(b) *Swimmer-Corridor* (evaluation of trajectory clips)

Figure 3: Results in the (a) *Driver* and (b) *Swimmer-Corridor* environments (shown on the left). We show the regret of a policy trained on the reward model compared to a policy trained on the true reward function as a function of the number of queries (middle plot). We report the cosine similarity between the learned and the true reward function (right plot). The plots show mean and standard error over 30 random seeds. IDRL finds significantly better policies, while not necessarily learning an overall more accurate model of the reward function. A similar plot for *Ant-Corridor* can be found in Appendix F.

We use *augmented random search* (Mania et al., 2018) as RL algorithm. For the *Swimmer-Corridor* we learn a linear policy, and for the *Ant-Corridor* we learn a hierarchical policy on top of pre-trained policies moving in four different directions. To generate candidate queries, we use a fixed, noisy exploration policy that moves along the whole corridor. Unfortunately, EPD is too expensive to evaluate in this environment and MR is not suited to this kind of queries.

**Results.** Figure 3b shows that IDRL needs significantly fewer queries to find a good policy than any of the baselines. IDRL adapts its queries to the policies that the current reward model induces: it initially samples clips in which the robot moves close to its starting position and shifts its focus to other regions as the reward model improves, and the learned policy starts to move. In contrast, the baselines make queries in the whole reachable space similarly often, and, therefore, waste queries in regions that are not directly relevant for improving the policy.

The computationally most expensive part of this implementation of IDRL is updating the candidate policies in each iteration. Updating them less often reduces the computational cost at the expense of potentially reducing the sample efficiency. Table 2 studies this trade-off and shows that IDRL outperforms the baselines even when the policies are updated only once at the beginning of training. In this extreme case, we reduce IDRL's runtime from about 40 hours to about 20 hours in *Swimmer-Corridor*, and from about 40 hours to about 10 hours in *Ant-Corridor*. This shows the benefits are larger when solving the RL problem is more expensive. Nonetheless, the baseline algorithms are still faster, and run for only 2−3 hours. This is because they do not require the additional inference steps necessary to optimize Equation (2). These results indicate that IDRL using full Thompson sampling to generate candidate policies can trade-off computational cost and sample efficiency, which allows it to be applied to large environments.

### 7.5    Can IDRL be scaled to a Deep RL setting?

Finally, we consider the Deep RL implementation of IDRL from Section 6, using the Soft Actor-Critic algorithm (SAC; Haarnoja et al., 2018). We test it on standard MuJoCo locomotion tasks, which are harder to learn with a DNN than with a GP model because the former encodes less prior information.

**Setup.** We consider a suite of standard tasks in MuJoCo implemented in OpenAI Gym (Brockman et al., 2016): *HalfCheetah-v3*, *Walker2d-v3*, *Hopper-v3*, *Ant-v3*, *Swimmer-v3*, *InvertedPendulum-v2*, *InvertedDoublePendulum-v2*, *Reacher-v2*. Similar to Christiano et al. (2017), we modify some environments to remove the termination conditions. Our environments differ slightly from Christiano et al., for the details see Appendix E.3. Our evaluation metric is a normalized score, averaged over all environments. A score of 0 corresponds to a random policy and a score of 100 is the performance of a policy trained on the true reward function. We provide results for the individual environments in Appendix F. Since IDRL tracks the candidate policies, it generates the candidate queries rolling

out the currently optimal policy *and* the candidate policies. However, the baselines do not have access to the candidate policies, and therefore consider a smaller set of potential queries. For a fair comparison, we perform an ablation where IDRL does not consider the candidate policies to generate candidate queries. Since IDRL maintains 3 (additional) candidate policies, it is roughly 4 times slower (about 80 hours on a single GPU) than the baselines (about 20 hours on a single GPU).

**Results.** Figure 4 shows that IDRL on average learns good policies significantly faster than the baselines. The individual results in each environment (in Appendix F) are more nuanced. IDRL clearly outperforms the baselines in some environments (e.g., *Hopper-v3*), performs comparable in other environments (e.g., *Walker2d-v3*), and performs worse than uniform sampling in a few environments (e.g., *HalfCheetah-v3*). Also while mostly using the candidate policy rollouts improves the performance of IDRL, this is not always the case (e.g., in *Swimmer-v3* the ablation performs better). This indicates that much of the variance might be caused by which queries are considered, which could be improved by using other exploration strategies than the candidate policies to generate candidate queries. Crucially, these experiments demonstrate that IDRL is scalable to high-dimensional, complex tasks, while still improving sample efficiency over existing methods for such tasks.

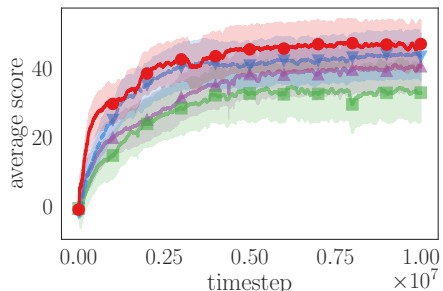

Figure 4: Normalized score of policies learned from 1400 (synthetic) comparisons of clips of the agent's behavior, averaged over all MuJoCo environments (higher is better). We show the mean and standard error of the score averaged over 5 random seeds per environment. The plot compares IDRL (red circles) to IGR (purple triangles) and uniform sampling (green squares), as well as an ablation of IDRL that does not use the candidate policies to generate additional candidate queries (blue triangles). EPD is too expensive and MR is not suited to this kind of queries.

## 8 Conclusion

We studied the problem of actively learning reward function models using as few expert queries as possible. We introduced *Information Directed Reward Learning* (IDRL), a novel information-theoretic algorithm that focuses on learning a good policy rather than attaining a low approximation error of the reward and that, differently from most prior methods, works with multiple types of feedback. We show it needs significantly fewer queries than prior methods and that it scales to complex environments.

**Limitations and future work.** The main practical limitation of IDRL is its computational cost. We demonstrated how to scale IDRL to complex environments, increasing the runtime by only a constant factor. While IDRL is still more demanding than most existing algorithms, it is preferable in situations where better sample efficiency is more important than low computational cost.

Our problem setup also has some conceptual limitations. We assume that interactions with the environment are cheap, which is not the case in many applications. Future work could aim to achieve low sample complexity in terms of environment interactions as well as reward queries. Moreover, we assume that the goal of RL is to learn a good policy in a single environment, which does not consider the problem of generalizing to other environment. In fact, being designed to learn a good policy in a single environment, IDRL might not be best for learning a reward model that generalizes well. To address this, future versions of IDRL could aim to learn a reward model that leads to good policies over a *distribution* of environments instead of a single environment.

Overall, we consider IDRL an addition to the set of existing active reward learning algorithms rather than a replacement of existing methods.

**Broader impact.** IDRL improves the sample efficiency of learning reward models, which is a step towards making RL a viable solution for real-world problems. RL systems can be used in various ways, and they could cause risks from malicious actors (Brundage et al., 2018). However, overall, learning reward models is likely to help in making RL more robust and safe (Leike et al., 2018).

Improving the sample efficiency of learning reward models, is crucial for making RL more useful. By addressing this problem, IDRL takes a step towards making RL a more viable solution for real-world problems.

## Acknowledgements

This research was supported through the Microsoft Swiss Joint Research Center. We thank Johannes Kirschner and Jonas Rothfuss for valuable feedback on an earlier version of this paper, and Nils Wilde for valuable comments about the Maximum Regret approach to reward learning.

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
