# A Proofs of propositions

In this section, we provide proofs of all results mentioned in the main paper. The results generally follow from well-known facts about Gaussian distributions and information theory.

**Proposition A.1.** If $\hat{r}(s)$ is a GP, the difference in expected return between two fixed policies $\pi, \pi'$ follows a Gaussian distribution $\mathcal{N}(\mu, \sigma^2)$ with

$$\mu = \mathbb{E}[\hat{G}(\pi) - \hat{G}(\pi')] = \langle \mathbf{v}_{\pi,\pi'}, \mu_{\hat{\mathbf{r}}} \rangle$$
$$\sigma^2 = \text{Var}[\hat{G}(\pi) - \hat{G}(\pi')] = \mathbf{v}_{\pi,\pi'}^T \cdot \Sigma_{\hat{\mathbf{r}}} \cdot \mathbf{v}_{\pi,\pi'}$$

where $\mathbf{v}_{\pi,\pi'} = \mathbf{f}^\pi - \mathbf{f}^{\pi'}$ is the difference between expected state-visitation frequencies of $\pi$ and $\pi'$ respectively, and $\mu_{\hat{\mathbf{r}}}$ and $\Sigma_{\hat{\mathbf{r}}}$ are the mean and covariance of the joint Gaussian distribution of the reward of all states $\pi$ or $\pi'$ visit.

*Proof.* If a random variable $X$ is Gaussian distributed $X \sim \mathcal{N}(\boldsymbol{\mu}, \Sigma)$, $\boldsymbol{\mu} \in \mathbb{R}^n, \Sigma \in \mathbb{R}^{n \times n}$, then for $\mathbf{a} \in \mathbb{R}^n$, $\mathbf{a}^T X$ is also Gaussian distributed $\mathbf{a}^T X \sim \mathcal{N}(\langle \mathbf{a}, \boldsymbol{\mu} \rangle, \mathbf{a}^T \Sigma \mathbf{a})$ (Wasserman, 2004, Theorem 14.2).

We can directly apply this fact to $\hat{\mathbf{r}} \sim \mathcal{N}(\mu_{\hat{\mathbf{r}}}, \Sigma_{\hat{\mathbf{r}}})$ and $\hat{G}(\pi) - \hat{G}(\pi') = \langle \mathbf{v}_{\pi,\pi'}, \hat{\mathbf{r}} \rangle$, resulting in

$$\hat{G}(\pi) - \hat{G}(\pi') \sim \mathcal{N}(\langle \mathbf{v}_{\pi,\pi'}, \mu_{\hat{\mathbf{r}}} \rangle, \mathbf{v}_{\pi,\pi'}^T \Sigma_{\hat{\mathbf{r}}} \mathbf{v}_{\pi,\pi'}).$$

$\square$

**Proposition 5.1.** If $\hat{r}(s)|\mathcal{D}$ is a GP, then $P(\hat{G}(\pi) - \hat{G}(\pi')|\mathcal{D})$ is Gaussian and:

$$\operatorname*{argmax}_{\pi,\pi' \in \Pi_c} H(\hat{G}(\pi) - \hat{G}(\pi')|\mathcal{D}) = \operatorname*{argmax}_{\pi,\pi' \in \Pi_c} \text{Var}[\hat{G}(\pi) - \hat{G}(\pi')|\mathcal{D}]$$
$$\operatorname*{argmax}_{q \in \mathcal{Q}_c} I(\hat{G}(\pi_1) - \hat{G}(\pi_2); (q, \hat{y})|\mathcal{D}) = \operatorname*{argmin}_{q \in \mathcal{Q}_c} \text{Var}[\hat{G}(\pi_1) - \hat{G}(\pi_2)|\mathcal{D} \cup \{(q, \hat{y})\}]$$

*Proof.* If a random variable $X$ is Gaussian distributed $X \sim \mathcal{N}(\mu, \sigma^2)$, then the entropy $H(X)$ is given by (Cover and Thomas, 2006, Theorem 8.4.1)

$$H(X) = \frac{1}{2}\log(2\pi e \sigma^2). \tag{3}$$

Proposition A.1 shows that the conditional distribution of $\hat{G}(\pi) - \hat{G}(\pi')|\mathcal{D}$ is Gaussian, which implies both statements.

For the first statement, observe that the entropy of $\hat{G}(\pi) - \hat{G}(\pi')|\mathcal{D}$ is

$$H(\hat{G}(\pi) - \hat{G}(\pi')|\mathcal{D}) = \frac{1}{2}\log(2\pi e \text{Var}[\hat{G}(\pi) - \hat{G}(\pi')|\mathcal{D}]), \tag{4}$$

and that two policies that maximize the variance on the r.h.s. also maximize the entropy, because the logarithm is a monotonic function.

To see the second statement, let $\hat{\Delta}_{\pi_1,\pi_2} = \hat{G}(\pi_1) - \hat{G}(\pi_2)$. Then

$$\operatorname*{argmax}_{q \in \mathcal{Q}_c} I(\hat{\Delta}_{\pi_1,\pi_2}; (q, \hat{y})|\mathcal{D})$$
$$= \operatorname*{argmax}_{q \in \mathcal{Q}_c} \left( H(\hat{\Delta}_{\pi_1,\pi_2}|\mathcal{D}) - H(\hat{\Delta}_{\pi_1,\pi_2}|\mathcal{D} \cup \{(q, \hat{y})\}) \right)$$
$$= \operatorname*{argmin}_{q \in \mathcal{Q}_c} H(\hat{\Delta}_{\pi_1,\pi_2}|\mathcal{D} \cup \{(q, \hat{y})\})$$
$$= \operatorname*{argmin}_{q \in \mathcal{Q}_c} \frac{1}{2}\log\left(2\pi e \text{Var}[\hat{\Delta}_{\pi_1,\pi_2}|\mathcal{D} \cup \{(q, \hat{y})\}]\right)$$
$$= \operatorname*{argmin}_{q \in \mathcal{Q}_c} \text{Var}[\hat{\Delta}_{\pi_1,\pi_2}|\mathcal{D} \cup \{(q, \hat{y})\}].$$

Here we wrote the information gain in terms of conditional entropies (Cover and Thomas, 2006, Theorem 2.4.1), and used that only one of the terms depends on $q$. This turns the maximization of information gain into a minimization of a conditional entropy. As before, we can further simplify this to minimizing conditional variance by using the entropy of a Gaussian and the fact that the logarithm is a monotonic function. $\square$

**Proposition 5.2.** Let $q$ be a linear reward query. If the prior belief about the reward $\hat{r}(s)$ is a GP, then the posterior belief about the reward $\hat{r}(s)|(q, y)$ is also a GP.

*Proof.* Let $q = (S, C)$ be a linear reward query, i.e., $S = \{s_1, \ldots, s_N\} \subseteq \mathcal{S}$ is a set of states and $C = \{c_1, \ldots, c_N\}$ a set of linear weights, and $y = \sum_{j=1}^{N} c_j r(s_j)$ the corresponding observation.

Let $S^* = \{s_1^*, \ldots, s_n^*\} \subseteq \mathcal{S}$ be a set of states for which we want to compute the posterior belief. We show that
$$P(\hat{r}(s_1^*), \ldots, \hat{r}(s_n^*)|(q, y)) \sim \mathcal{N}(\boldsymbol{\mu}_q^*, \Sigma_q^*)$$
for some $\boldsymbol{\mu}_q^*$ and $\Sigma_q^*$. Because this holds for any set of states $S^*$, it shows that the posterior reward model is a GP.

We define the following vector notation:
$$\mathbf{c} = (c_1, \ldots, c_N)^T \in \mathbb{R}^N$$
$$\hat{\mathbf{r}} = (\hat{r}(s_1), \ldots, \hat{r}(s_N))^T \in \mathbb{R}^N$$
$$\hat{\mathbf{r}}^* = (\hat{r}(s_1^*), \ldots, \hat{r}(s_n^*))^T \in \mathbb{R}^n$$
such that $y = \langle \mathbf{c}, \hat{\mathbf{r}} \rangle$.

The prior distribution of $\hat{\mathbf{r}}$ is Gaussian, i.e.,
$$P(\hat{\mathbf{r}}|S) \sim \mathcal{N}(\boldsymbol{\mu}, \Sigma),$$
with mean $\boldsymbol{\mu}$ and covariance $\Sigma$.

Because $\hat{y}$ is a linear function of $\hat{\mathbf{r}}$ plus Gaussian noise, the prior distribution of $\hat{y}$ is also Gaussian (Wasserman, 2004, Theorem 14.2):
$$P(\hat{y}|S, C) \sim \mathcal{N}(\langle \mathbf{c}, \boldsymbol{\mu} \rangle, \mathbf{c}^T \Sigma \mathbf{c} + \sigma_n^2 I).$$

Further, $\hat{\mathbf{r}}$ and $\hat{\mathbf{r}}^*$ are jointly Gaussian distributed:
$$P\left(\left[\begin{smallmatrix} \hat{\mathbf{r}} \\ \hat{\mathbf{r}}^* \end{smallmatrix}\right] | S, S^*\right) \sim \mathcal{N}\left(\left[\begin{smallmatrix} \boldsymbol{\mu} \\ \boldsymbol{\mu}^* \end{smallmatrix}\right], \left[\begin{smallmatrix} \Sigma & \Sigma^* \\ (\Sigma^*)^T & \Sigma^{**} \end{smallmatrix}\right]\right)$$
where $\boldsymbol{\mu}^*$ is the mean of $\hat{\mathbf{r}}^*$, and $\Sigma^* = \text{Cov}[\hat{\mathbf{r}}, \hat{\mathbf{r}}^*]$ and $\Sigma^{**} = \text{Cov}[\hat{\mathbf{r}}, \hat{\mathbf{r}}^*]$ denote the components of the joint covariance matrix.

Hence, $\hat{\mathbf{r}}^*$ and $\hat{y}$ are also jointly Gaussian distributed:
$$P\left(\left[\begin{smallmatrix} \hat{\mathbf{r}}^* \\ \hat{y} \end{smallmatrix}\right] | S, C, S^*\right) \sim \mathcal{N}\left(\left[\begin{smallmatrix} \boldsymbol{\mu}^* \\ \langle \mathbf{c}, \boldsymbol{\mu} \rangle \end{smallmatrix}\right], \left[\begin{smallmatrix} \Sigma^{**} & (\Sigma^*)^T \mathbf{c} \\ \mathbf{c}^T \Sigma^* & \mathbf{c}^T \Sigma \mathbf{c} + \sigma_n^2 I \end{smallmatrix}\right]\right)$$
where we used the linearity of the covariance function to find the covariance matrix:
$$\text{Cov}[\hat{y}, \hat{\mathbf{r}}^*] = \text{Cov}[\langle \mathbf{c}, \hat{\mathbf{r}} \rangle, \hat{\mathbf{r}}^*]$$
$$= \mathbf{c}^T \text{Cov}[\hat{\mathbf{r}}, \hat{\mathbf{r}}^*] = \mathbf{c}^T \Sigma^*$$
$$\text{Cov}[\hat{\mathbf{r}}^*, \hat{y}] = (\Sigma^*)^T \mathbf{c}$$

Finally, we can use standard results on conditioning Gaussian distributions (cf. Rasmussen and Williams, 2006, Chapter A.2) to find that the conditional distribution is still Gaussian:
$$P(\hat{\mathbf{r}}^*|(q, y)) = P(\hat{\mathbf{r}}^*|y, S, C, S^*) \sim \mathcal{N}(\boldsymbol{\mu}_q^*, \Sigma_q^*)$$
with
$$\boldsymbol{\mu}_q^* = \boldsymbol{\mu}^* + ((\Sigma^*)^T \mathbf{c})(\mathbf{c}^T \Sigma \mathbf{c} + \sigma_n^2 I)^{-1}(y - \langle \mathbf{c}, \boldsymbol{\mu} \rangle)$$
$$\Sigma_q^* = \Sigma^{**} - ((\Sigma^*)^T \mathbf{c})(\mathbf{c}^T \Sigma \mathbf{c} + \sigma_n^2 I)^{-1}(\mathbf{c}^T \Sigma_*).$$

When conditioning the distribution, we replaced our belief about the observation $\hat{y}$ with its actual realization $y$. $\square$

# B    Linear reward queries

We consider linear reward queries for our implementation of IDRL with GP models, which makes all computations analytically tractable (cf. Section 5). Linear reward queries can be used to model many different observation types that are typical in practical settings. In this section, we recall the definition of linear reward queries, and then present a few particularly common types of linear reward queries, which include all query types used in our empirical evaluation of IDRL with a GP model.

**Definition 5.1.** We call $q = (S, C)$ a *linear reward query*, if it consists of states $S = \{s_1, \ldots, s_N\}$ and linear weights $C = \{c_1, \ldots, c_N\}$, and the response to query $q$ is a linear combination of rewards $y = \sum_{j=1}^{N} c_j r(s_j) + \varepsilon$, with Gaussian noise $\varepsilon \sim \mathcal{N}(0, \sigma_n^2)$.

**Single state rewards.** If $N = 1$, a query consists of a single state $q_i = s_i \in \mathcal{S}$, for which the expert provides a noisy reward $y = r(s_i) + \varepsilon$.

**Return of trajectories.** For $N > 1$ and all $c_i = 1$, the agent observes the sum of rewards of multiple states. The set $S$ could, e.g., contain the states in a trajectory or a sub-sequence of it. Then, the queries ask about the return, i.e., sum of rewards, of this sequence or states.

**Comparisons of states and trajectories.** We can model a comparison of the reward in states $s_a$ and $s_b$ by defining $S = \{s_a, s_b\}$ and defining $C = \{1, -1\}$. Then the agent might observe $y = r(s_a) - r(s_b)$. In practice, comparison queries usually result in binary feedback, i.e., the expert states that either $s_a$ or $s_b$ is preferred. We can model this, e.g., with a Bernoulli distribution $P(y = 1) = (1 + r(s_a) - r(s_b))/2$, if all rewards are between 0 and 1. The observations from this distribution have expectation $r(s_a) - r(s_b)$ and the noise model is subgaussian, which we can approximate with a Gaussian noise distribution (cf. Kirschner et al., 2020). Hence, we can model such comparison queries as linear reward queries. We can model comparisons between two sets of states, e.g., between two trajectories, analogously. Other observation models for comparisons have been proposed in the literature, such as softmax (Sadigh et al., 2017), probit (Bıyık et al., 2020a) or Bernoulli distributions with constant probability (Wilde et al., 2020). While we focus on linear observations, IDRL could be extended to these alternatives by using approximate inference to update the reward model, similar to Bıyık et al. (2020a).

# C    Connection to multi-armed bandits

In the main paper, we motivated IDRL from information-theoretic considerations. However, there are close connections to related algorithms in multi-armed bandits (MAB) that can serve as additional motivation.

Efficient exploration is extensively studied in MAB problems (Bubeck and Cesa-Bianchi, 2012). Recent work successfully uses decision criteria based on information gain in various MAB problems (Russo and Van Roy, 2014). However, our setting is no standard MAB problem, because we do not directly observe the quantity we are optimizing for, i.e., the return of a policy.

In this section, we discuss two settings that are more closely related to our setting: the *linear partial monitoring* problem and *transductive linear bandits*.

## C.1    Linear partial monitoring

Our setting is closely related to partial monitoring problems, which generalize the standard MAB to cases where the agent's observations provide only indirect information about the reward (Rustichini, 1999). For tabular MDPs, our setting can be interpreted as a linear partial monitoring problem. Let $\mathbf{r}$ be a vector of all rewards in a tabular MDP. We consider observations that are a linear function of the rewards $r(s_i) = \langle \mathbf{c}, \mathbf{r} \rangle$, and the optimization target is also a linear function of the reward vector $G(\pi) = \langle \mathbf{f}^\pi, \mathbf{r} \rangle$. Kirschner et al. (2020) analyze linear partial monitoring problems and propose an information gain based criterion for selecting observations. One criterion they propose to measure information gain, called *directed information gain*, is equivalent to our information gain criterion (Kirschner et al., 2020, App. B.2). However, they consider cumulative regret minimization, and, therefore, their algorithm has to trade-off the information gain of an observation with its expected regret. In our setting, minimizing cumulative regret would correspond to maximizing $\sum_{t=1}^{T} G(\bar{\pi}_t)$, where $\bar{\pi}_t$ is the policy that IDRL returns if it is stopped after $t$ iterations. Instead,

we just evaluate the final policy and aim to maximize $G(\bar{\pi}_T)$. Consequently, our algorithm directly uses directed information gain as a selection criterion.

## C.2 Transductive linear bandits

Our setting is a *pure exploration* problem (Bubeck et al., 2009): we only evaluate the performance of the final policy after a fixed budget of queries and not the intermediary policies. Our problem is closely related to pure exploration in *transductive linear bandits* that consider maximizing a linear reward function in a set $\mathcal{Z}$ by making queries in a potentially different set $\mathcal{X}$ (Fiez et al., 2019). In fact, for a tabular MDP, our problem is a special case of the transductive linear bandit setting. Moreover, we can understand IDRL as an adaptive version of the RAGE algorithm introduced by Fiez et al. (2019).

To see the connection between both settings, let us first define the transductive linear bandit problem.

**Definition C.1** (Fiez et al., 2019). A *transductive linear bandit problem* is defined by two sets $\mathcal{X} \subset \mathbb{R}^d$ and $\mathcal{Z} \subset \mathbb{R}^d$, where the goal is to find $\mathrm{argmax}_{\mathbf{z} \in \mathcal{Z}} \langle \mathbf{z}, \boldsymbol{\theta}^* \rangle$ for some hidden parameter vector $\boldsymbol{\theta}^* \in \mathbb{R}^d$. However, instead of observing this objective directly, the learning agent interacts with the bandit at each time-step by selecting an arm $\mathbf{x} \in \mathcal{X}$ to play, and then observing $\langle \mathbf{x}, \boldsymbol{\theta}^* \rangle + \eta$ where $\eta$ is independent, zero-mean, subgaussian noise. The agent's goal is to find the maximum in $\mathcal{Z}$ by making as few queries in $\mathcal{X}$ as possible.

**Proposition C.1.** For finite state and action spaces, a fixed set of candidate policies $\Pi_c$ and a set of linear reward queries $\mathcal{Q}_c$, our reward learning problem is a transductive linear bandit problem with $\mathcal{Z} = \{\mathbf{f}^\pi | \pi \in \Pi_c\} \subset \mathbb{R}^{|\mathcal{S}|}$ and $\mathcal{X} = \{\mathbf{c}_i | i \in \{1, \ldots, |\mathcal{Q}_c|\}\} \subset \mathbb{R}^{|\mathcal{S}|}$ a set of linear observations.

To see this, note that our goal is to maximize $G(\pi) = \langle \mathbf{f}^\pi, \mathbf{r} \rangle$ and we query linear combinations of rewards in each round $\langle \mathbf{c}_{i_t}, \mathbf{r} \rangle$. Here $i_t$ is the index of the query that the agent selects at time $t$, and $\mathbf{c}_{i_t}$ is a vector of linear weights that defines query $q_{i_t}$.

To understand the connection between IDRL and the RAGE algorithm proposed by Fiez et al. (2019), it is helpful to assume that the reward function is a linear function of some features of the state, and to use a linear kernel for the GP model, which is equivalent to Bayesian linear regression.

Let $\boldsymbol{\phi} : \mathcal{S} \to \mathbb{R}^d$ be a feature function, and the true reward function $r(s) = \langle \boldsymbol{\phi}(s), \boldsymbol{\theta}^* \rangle$. Similarly, we can define a feature vector for each query $q \in \mathcal{Q}_c$ and overload the notation $\boldsymbol{\phi}(q) = \sum_{i=1}^{N} C_i \boldsymbol{\phi}(s_i)$. Also, we can write the expected return of a policy as $G(\pi) = \langle \mathbf{f}_\phi^\pi, \boldsymbol{\theta}^* \rangle$ with $\mathbf{f}_\phi^\pi = (\mathbf{f}^\pi)^T \Phi$ and $\Phi = (\boldsymbol{\phi}(s_1), \ldots, \boldsymbol{\phi}(s_{|\mathcal{S}|}))^T$.

To solve the transductive linear bandit problem, the RAGE algorithm proceeds in multiple rounds, in each of which it follows an allocation rule

$$\lambda_t^* = \underset{\lambda \in \Delta_\mathcal{X}}{\mathrm{argmin}} \max_{\pi_1, \pi_2 \in \widehat{\mathcal{Z}}_t} \|\mathbf{f}_\phi^{\pi_1} - \mathbf{f}_\phi^{\pi_2}\|_{A_\lambda^{-1}}^2 \tag{5}$$

where $A_\lambda = \sum_{q_i \in \mathcal{Q}_c} \lambda_i \boldsymbol{\phi}(q_i) \boldsymbol{\phi}(q_i)^T$, and where $\Delta_\mathcal{X}$ is the probability simplex over candidate queries, so this rule would select query $q_i$ at round $t$ with probability $(\lambda_t^*)_i$. Additionally, RAGE keeps track of a set of plausibly optimal arms $\widehat{\mathcal{Z}}_t$, i.e., plausibly optimal policies in our case. RAGE ensures that the suboptimality gap of arms in this set shrinks exponentially as the algorithm proceeds.

The next proposition provides an alternative notation for IDRL that shows a formal similarity to RAGE.

**Proposition C.2.** Assume we estimate $\hat{\boldsymbol{\theta}}$ with Bayesian linear regression with noise variance $\sigma^2$, and prior $\hat{\boldsymbol{\theta}} \sim \mathcal{N}(0, \alpha^{-1}I)$ after collecting data

$$\mathcal{D} = ((\boldsymbol{\phi}(q_{i_1}), y_{i_1}), \ldots, (\boldsymbol{\phi}(q_{i_{t-1}}), y_{i_{t-1}})).$$

Also, assume an infinitely wide prior $\alpha^{-1} \to \infty$.

We can then write the maximization in the first step of IDRL as

$$\underset{\pi, \pi' \in \Pi_c}{\mathrm{argmax}} \, H(\hat{G}(\pi) - \hat{G}(\pi')|\mathcal{D}) = \underset{\pi, \pi' \in \Pi_c}{\mathrm{argmax}} \|\mathbf{f}_\phi^\pi - \mathbf{f}_\phi^{\pi'}\|_{A_\mathcal{D}^{-1}}^2$$

where $A_\mathcal{D} = \sum_{q_i \in \mathcal{Q}_c} N_i \boldsymbol{\phi}(q_i) \boldsymbol{\phi}(q_i)^T$.

Furthermore, for a given pair of policies, $\pi_1$ and $\pi_2$, we can write the maximization in the second step of IDRL as

$$\underset{q \in \mathcal{Q}_c}{\operatorname{argmax}} \, I(\hat{G}(\pi_1) - \hat{G}(\pi_2); (q, \hat{y})|\mathcal{D})$$

$$= \underset{q \in \mathcal{Q}_c}{\operatorname{argmin}} \, \|\mathbf{f}_\phi^{\pi_1} - \mathbf{f}_\phi^{\pi_2}\|_{A_{\mathcal{D},q}^{-1}}^2$$

where $A_{\mathcal{D},q} = \phi(q)\phi(q)^T + \sum_{q_i \in \mathcal{Q}_c} N_i \phi(q_i)\phi(q_i)^T$ and $N_i$ is the number of times $q_i$ occurs in $\mathcal{D}$.

*Proof.* In the Bayesian linear regression setting (cf. Bishop, 2006, Chapter 3.3) with prior weight distribution $\mathbf{w} \sim \mathcal{N}(\mathbf{0}, \alpha^{-1})$, the posterior weight distribution is a Gaussian with covariance matrix

$$\Sigma_\theta = \left( \alpha I + \sigma^{-2} \sum_{q \in \mathcal{D}} \phi(q)\phi(q)^T \right)^{-1}$$

$$= \left( \alpha I + \sigma^{-2} A_{\mathcal{D}} \right)^{-1}$$

$$A_{\mathcal{D}} = \sum_{q \in \mathcal{D}} \phi(q)\phi(q)^T$$

$$= \sum_{q_i \in \mathcal{Q}_c} N_i \phi(q_i)\phi(q_i)^T$$

For an infinitely wide prior ($\alpha^{-1} \to \infty$): $\Sigma_\theta \to \sigma^2 A_{\mathcal{D}}^{-1}$.

Using the linear mapping from $\hat{\boldsymbol{\theta}}$ to the expected return of a policy $\hat{G}(\pi)$, the posterior variance of the difference in return between two policies is

$$\operatorname{Var}[\hat{G}(\pi_1) - \hat{G}(\pi_2)|\mathcal{D}] = \sigma^2 (\mathbf{v}_{\pi_1, \pi_2}^\phi)^T A_{\mathcal{D}}^{-1} \mathbf{v}_{\pi_1, \pi_2}^\phi \tag{6}$$

where $\mathbf{v}_{\pi_1, \pi_2}^\phi = \mathbf{f}_\phi^{\pi_1} - \mathbf{f}_\phi^{\pi_2}$

The first part of the statement follows using Proposition 5.1:

$$\underset{\pi, \pi' \in \Pi_c}{\operatorname{argmax}} \, H(\hat{G}(\pi) - \hat{G}(\pi')|\mathcal{D})$$

$$= \underset{\pi, \pi' \in \Pi_c}{\operatorname{argmax}} \, \operatorname{Var}[\hat{G}(\pi) - \hat{G}(\pi')|\mathcal{D}] \qquad \text{(Proposition 5.1)}$$

$$= \underset{\pi, \pi' \in \Pi_c}{\operatorname{argmax}} \, \sigma^2 (\mathbf{v}_{\pi, \pi'}^\phi)^T A_{\mathcal{D}}^{-1} \mathbf{v}_{\pi, \pi'}^\phi \qquad \text{(Equation (6))}$$

$$= \underset{\pi, \pi' \in \Pi_c}{\operatorname{argmax}} \, (\mathbf{v}_{\pi, \pi'}^\phi)^T A_{\mathcal{D}}^{-1} \mathbf{v}_{\pi, \pi'}^\phi$$

$$= \underset{\pi, \pi' \in \Pi_c}{\operatorname{argmax}} \, \|\mathbf{v}_{\pi, \pi'}^\phi\|_{A_{\mathcal{D}}^{-1}}$$

$$= \underset{\pi, \pi' \in \Pi_c}{\operatorname{argmax}} \, \|\mathbf{f}_\phi^\pi - \mathbf{f}_\phi^{\pi'}\|_{A_{\mathcal{D}}^{-1}}$$

After defining

$$A_{\mathcal{D},q} = A_{\mathcal{D} \cup \{(q,y)\}}$$

$$= \phi(q)\phi(q)^T + \sum_{q_i \in \mathcal{Q}_c} N_i \phi(q_i)\phi(q_i)^T,$$

the second part of the statement follows analogously to the first one after applying Proposition 5.1:

$$\underset{q \in \mathcal{Q}_c}{\mathrm{argmax}}\, I(\hat{G}(\pi_1) - \hat{G}(\pi_2); (q, \hat{y})|\mathcal{D})$$

$$= \underset{q \in \mathcal{Q}_c}{\mathrm{argmin}}\, \mathrm{Var}[\hat{G}(\pi_1) - \hat{G}(\pi_2)|\mathcal{D} \cup \{q, y\}] \qquad (\text{Prop. } 5.1)$$

$$= \underset{q \in \mathcal{Q}_c}{\mathrm{argmin}}\, \sigma^2 (\mathbf{v}_{\pi_1, \pi_2}^{\phi})^T A_{\mathcal{D}, q}^{-1} \mathbf{v}_{\pi_1, \pi_2}^{\phi} \qquad (\text{Equation } (6))$$

$$= \underset{q \in \mathcal{Q}_c}{\mathrm{argmin}}\, (\mathbf{v}_{\pi_1, \pi_2}^{\phi})^T A_{\mathcal{D}, q}^{-1} \mathbf{v}_{\pi_1, \pi_2}^{\phi}$$

$$= \underset{q \in \mathcal{Q}_c}{\mathrm{argmin}}\, \|\mathbf{v}_{\pi_1, \pi_2}^{\phi}\|_{A_{\mathcal{D}, q}^{-1}}$$

$$= \underset{q \in \mathcal{Q}_c}{\mathrm{argmin}}\, \|\mathbf{f}_{\phi}^{\pi_1} - \mathbf{f}_{\phi}^{\pi_2}\|_{A_{\mathcal{D}, q}^{-1}}$$

$\square$

Comparing this proposition with equation (5) shows a formal similarity between both algorithms. In particular, we can understand IDRL as a version of equation (5) that adapts to the data seen so far and selects the next observation that would minimize this objective. Instead of the matrix $A_\lambda$ that is induced by the allocation rule $\lambda$, IDRL computes the variances using $A_\mathcal{D}$, i.e., based on data observed in the past, and using $A_{\mathcal{D},q}$, i.e., evaluating the effect of an additional observation. Additionally, IDRL performs two separate optimizations which one can consider as an approximation to the min-max problem in equation (5), which would be infeasible to evaluate in our setting. Similar to RAGE, IDRL keeps track of a set of plausibly optimal policies. However, IDRL uses Thompson sampling, while RAGE uses suboptimality gaps to build this set.

## D   Implementation details of IDRL with GP reward models

In this section we describe our implementation of IDRL, the baselines we compare to, and our environments in more detail. For the choice of hyperparameters and additional implementation details we refer to the code of our experiments.

### D.1   Thompson sampling

For some environments, a set of potentially optimal policies $\Pi_c$ might be available. In other cases, we use Thompson sampling (TS) to generate $\Pi_c$. Our TS approach is shown in Algorithm 2. We select $N$ policies by sampling reward functions from the posterior belief of the reward model and then finding optimal policies for them using some RL algorithm (RL in the pseudocode).

The set of candidate policies $\Pi_c$ should be updated regularly during IDRL to reflect the current posterior belief on optimal policies. In our experiments, we use $N = 5$ policies, and update them in each iteration, if not stated differently in the text.

Depending on the environment, we use different RL algorithms. For *Chain*, *Junction*, and *Gridworld* environments, we use an exact solver (using linear programming, or a lookup table of all deterministic policies). For the *Driver* environment, we use the L-BFGS-B solver provided by Sadigh et al. (2017). However, we combine it with a lookup table of pre-computed policies to reduce noise, as suggested by Wilde et al. (2020). Whenever the lookup table contains a policy that is better than the one returned by the solver, we use the policy from the table instead. In the *MuJoCo* environment, we use *augmented random search* with linear policies (Mania et al., 2018) as RL.

### D.2   Details on the baselines

In this section, we discuss the baselines in more detail. In Algorithm 3 we present pseudocode for the general reward learning algorithm that all of our baselines implement. They only differ in the choice of acquisition function in line 5. In the following, we discuss the different choices.

#### D.2.1   Uniform sampling

The uniform sampling baseline runs Algorithm 3 with $q^*$ sampled uniformly from $\mathcal{Q}_c$ instead of line 5.

**Algorithm 2** Thompson sampling for creating a set of candidate policies $\Pi_c$.

$\Pi_c \leftarrow \{\}$
**for** $i \in \{1, \ldots, N\}$ **do**
     sample reward function $r_i \sim P(\hat{r}|\mathcal{D})$
     $\pi_i^* \leftarrow \mathrm{RL}(r_i)$
     $\Pi_c \leftarrow \Pi_c \cup \{\pi_i^*\}$
**end for**
**return** $\Pi_c$

---

**Algorithm 3** Generic reward learning algorithm using an acquisition function $u(q, \mathcal{D})$. Our baselines use information gain and expected improvement for $u$. Uniform sampling samples $q^*$ uniformly from $\mathcal{Q}_c$ instead of line 5.

1: $\mathcal{D} \leftarrow \{\}$
2: Initialize reward model with prior distribution $P(\hat{r})$
3: **while** not converged **do**
4:      Select a query:
5:          $q^* \in \mathrm{argmax}_{q \in \mathcal{Q}_c} u(q, \mathcal{D})$
6:      Query $q^*$ and update reward model:
7:          $y^* \leftarrow$ Response to query $q^*$
8:          $P(\hat{r}|\mathcal{D} \cup \{(q^*, y^*)\}) \propto P(y^*|\hat{r}, \mathcal{D}, q^*)P(\hat{r}|\mathcal{D})$
9:          $\mathcal{D} \leftarrow \mathcal{D} \cup \{(q^*, y^*)\}$
10: **end while**
11: $\bar{r} \leftarrow$ mean estimate of the reward model
12: $\bar{\pi}^* \leftarrow \mathrm{RL}(\bar{r})$
13: **return** $\bar{\pi}^*$

### D.2.2 Information gain on the reward

Another baseline uses information gain on the reward as acquisition function for Algorithm 3, that is $u(q, \mathcal{D}) = I((q, \hat{y}); \hat{r}|\mathcal{D})$. Note that we can write this information gain in terms of conditional entropies (Cover and Thomas, 2006, Theorem 2.4.1):

$$I((q, \hat{y}); \hat{r}|\mathcal{D}) = H(\hat{y}|\mathcal{D}, q) - H(\hat{y}|\hat{r}, \mathcal{D}, q). \tag{7}$$

If we assume that $q$ is a linear reward query, it is described by a set of states $S = \{s_1, \ldots, s_n\}$ and a set of linear weights $C = \{c_1, \ldots, c_n\}$, such that $\hat{y} = \sum_{i=1}^{n} c_i \hat{r}(s_i) + \varepsilon$. Then, the second term of equation (7) is constant because $\hat{r}$ contains all information about $\hat{y}$. Further, the distribution $P(\hat{y}|\mathcal{D}, q)$ is Gaussian, and its entropy is (Cover and Thomas, 2006, Theorem 8.4.1):

$$H(\hat{y}|\mathcal{D}, q) = \frac{1}{2} \log \left( 2\pi e \mathrm{Var}[\hat{y}|\mathcal{D}, q] \right).$$

Because the logarithm is a monotonic function, we can show, analogously to Proposition 5.1, that

$$\underset{q \in \mathcal{Q}_c}{\mathrm{argmax}} \, I((q, \hat{y}); \hat{r}|\mathcal{D}) = \underset{q \in \mathcal{Q}_c}{\mathrm{argmax}} \, \mathrm{Var}[\hat{y}|\mathcal{D}, q].$$

Hence, for a GP reward model with linear reward observations, using $u(q, \mathcal{D}) = I((q, \hat{y}); \hat{r}|\mathcal{D})$ is equivalent to using $u(s, \mathcal{D}) = \mathrm{Var}[\hat{y}|\mathcal{D}, q]$, which is what we do in practice.

### D.2.3 Expected improvement

We define the *expected improvement* (EI) acquisition function as:

$$u(s, \mathcal{D}) = L \cdot \Phi(M) + \mathrm{Var}[\hat{y}|\mathcal{D}, q] \cdot \rho(M),$$

$$M = \frac{L}{\mathrm{Var}[\hat{y}|\mathcal{D}, q]},$$

$$L = \mathbb{E}[\hat{y}|\mathcal{D}, q] - y_{\max} - \xi,$$

where $\rho$ and $\Phi$ are the probability density function, and the cumulative density function of the standard normal distribution, respectively. $y_{\max}$ is the highest observation made so far. If $\mathrm{Var}[\hat{y}|\mathcal{D}, q] = 0$

**Algorithm 4** The expected policy divergence (EPD) acquisition function, introduced by Daniel et al. (2015), adapted to our setting. The algorithm computes EPD for a given query $q$.

---

$\bar{r} \leftarrow$ mean estimate of the model $P(\hat{r}|\mathcal{D})$
$\tilde{\pi} \leftarrow \texttt{RL}(\bar{r})$
$\tilde{y} \leftarrow \mathbb{E}[\hat{y}|\mathcal{D}, q] + \text{Var}[\hat{y}|\mathcal{D}, q]$
$P(\hat{r}|\mathcal{D} \cup \{(q, \tilde{y})\}) \propto P(\tilde{y}|\hat{r}, \mathcal{D}, q)P(\hat{r}|\mathcal{D})$
$\bar{r}^* \leftarrow$ mean estimate of the model $P(\hat{r}|\mathcal{D} \cup \{(q, \tilde{y})\})$
$\pi^* \leftarrow \texttt{RL}(\bar{r}^*)$
$u(q, \mathcal{D}) \leftarrow d(\tilde{\pi}, \pi^*)$
**return** $u(q, \mathcal{D})$

---

we define $u(s, \mathcal{D}) = 0$. $\xi$ is a hyperparameter, that we set to $0.001$. EI quantifies how much higher a new observation is expected to be than the highest observation made so far. In our setting, EI is only applicable if observations are numerically, e.g., if the observations are rewards of individual states or trajectories. In particular, we cannot use EI if the queries are comparisons of states or trajectories.

### D.2.4 Expected policy divergence

Daniel et al. (2015) introduce the expected policy divergence (EPD) acquisition function. EPD compares two policies: $\tilde{\pi}$ is trained from a reward model conditioned on the current dataset $\mathcal{D}$, and $\pi^*$ estimates a policy trained from a reward model conditioned on $\mathcal{D} \cup \{(q, y)\}$. EPD aims to quantify the effect of making a query $q$ and observing response $y$ on the currently optimal policy. For each potential query it assumes an observation at an upper confidence bound conditioned on the current model, and then selects observations that maximize some distance measure between policies $d(\tilde{\pi}, \pi^*)$. Algorithm 4 shows how EPD is computed in our setting, which our implementation combines with Algorithm 3. EPD requires solving an RL problem for each potential observation, which makes it computationally infeasible for bigger environments.

Daniel et al. (2015) introduce EPD using the KL divergence $D_{\text{KL}}(\tilde{\pi}\|\pi^*)$ as distance measure $d(\tilde{\pi}, \pi^*)$. However, in most of our experiments, the policies are deterministic, in which case the KL divergence is not well-defined. For tabular environments, we define $d$ to count the number of states in which the policies differ. For the *Driver* environment we use an $\ell_2$-distance between the policy representations.

### D.2.5 Maximum regret

Wilde et al. (2020) introduce the *Maximum Regret* (MR) acquisition function for selecting queries that compare two policies. They assume a set of candidate reward functions $\mathcal{R}_c = \{r_1, \ldots, r_n\}$ to be given and then consider a set of candidate policies $\Pi_c = \{\pi_1, \ldots, \pi_n\}$, where each policy $\pi_i$ is optimal for one of the reward functions $r_i$.

MR can be used to select comparison queries of the form $q = (\pi_i, \pi_j)$. In practice, we use MR for queries that compare trajectories sampled from $\pi_i$ and $\pi_j$.

MR aims to compare policies $\pi_i$ and $\pi_j$ that perform poorly when evaluated under each other's reward function $r_j$ and $r_i$ respectively. To formalize this in our notation, let us introduce the notation $G_{r_i}(\pi_j)$ to indicate the return of policy $\pi_j$ evaluated using reward function $r_i$. MR is defined as

$$u((\pi_i, \pi_j), \mathcal{D}) = P(r_i|\mathcal{D}) \cdot P(r_j|\mathcal{D}) \cdot (R(r_i, r_j) + R(r_j, r_i))$$

where $R(r_i, r_j)$ is a measure of regret of a policy optimized for reward function $r_i$ when evaluated under reward function $r_j$. Wilde et al. (2020) use a regret measure based on a ratio of returns: $R(r_i, r_j) = 1 - G_{r_j}(\pi_i)/G(\pi_j)$. However, this measure is only meaningful if all rewards are positive, which is not the case in our experiments. Therefore, we instead use a regret measure based on differences of returns: $R(r_i, r_j) = G(\pi_j) - G_{r_j}(\pi_i)$.[5]

---

[5] This change was suggested by the authors of Wilde et al. (2020) in personal communication to deal with negative rewards.

For computing the probabilities $P(r_i|\mathcal{D})$, Wilde et al. (2020) use a simple Bayesian model that assumes a uniform prior and a likelihood of making an observation of the form

$$P(y = +1|q = (\pi_i, \pi_j)) = \begin{cases} p & \text{if } G(\pi_i) > G(\pi_j) \\ 1 - p & \text{else} \end{cases} \tag{8}$$

with $0.5 < p \leq 1$. In the main paper we report results for MR using a GP reward model. We tested the simple Bayesian model in equation (8) in preliminary experiments, and found it to result in comparable results to the GP model with observations simulated using a linear observation model.

In our experiments, MR performed worse than reported by Wilde et al. (2020). This difference is likely explained by differences in the implementation of the reward model, the environment, or the acquisition function. Unfortunately, Wilde et al. (2020) did not publicly release code for their implementation of MR. Therefore, we were unable to reproduce their exact setup and results, and could not investigate differences between our implementation and theirs in detail.

### D.3   Details on the environments

This section provides more details on the environments we test IDRL with a GP model on. We start by introducing two additional environments: *Chain* and *Junction*, which are not presented in the main paper. Then, we discuss the *Gridworld*, *Driver*, and MuJoCo *Corridor* experiments from the main paper. Appendix F provides more detailed results for all environments.

#### D.3.1   Chain

Figure 5a shows the *Chain* environment. It has a discrete state space with $N$ states, and a discrete action space with 2 actions $a_r$ and $a_l$. In the first $M$ states of the chain both actions moves the agent right, whereas in the last $N - M$ states $a_l$ moves the agent left and $a_r$ moves the agent right. The dynamics are deterministic. The initial state distribution is uniform over the state space.

For the GP model of the reward, we choose a squared-exponential (SE) kernel

$$k(s, s') = \sigma^2 \exp\left(-\frac{d(s, s')^2}{2l^2}\right)$$

with variance $\sigma = 2$ and lengthscale $l = 3$. The distance $d$ counts the number of states between $s$ and $s'$ on the chain.

The *Chain* environment shows that, to select informative queries, it is important to consider in which states the agents actions change the transition probabilities. Queries about the first $M$ states are less informative, because in these states the agent cannot choose how to move.

#### D.3.2   Junction

Figure 5b shows the *Junction* environment. It has a discrete state space with $N + 2M$ states, and a discrete action space with 2 actions $a_1$ and $a_2$. In the first $N$ states either action moves the agent right. From state $s_N$ action $a_1$ moves the agent to $s_{A_1}$ and action $a_2$ moves the agent to $s_{B_1}$. In either of the two paths the agent moves to one of the adjacent states with probability $0.5$, independent of the action it took. The reward of states $s_1, \ldots, s_N$ is 0, the reward of states $s_{B_1}, \ldots, s_{B_M}$ is 0.8, and the reward of state $s_{A_i}$ is

$$r(s_{A_i}) = 1 - \left(0.7 \cdot \frac{i}{M} - 1\right)^2$$

This reward function ensures that the average reward in the upper chain is smaller than $0.8$ but the maximum reward is bigger than $0.8$. The initial state distribution is uniform over the state space.

For the GP model of the reward, we choose a SE kernel with variance $\sigma = 2$ and lengthscale $l = 3$. The distance $d$ measures the shortest path between $s$ and $s'$ on graph that defines the *Junction* (disregarding the transition function).

The *Junction* environment shows that it is not sufficient to select queries that are informative about the maximum of the reward function. Instead, it is important to consider the specifics of the environment to determine informative queries. In the *Junction* environment, the agent has to find the path with the higher average reward instead of the one with the higher maximum reward.

### D.3.3 Gridworld

The *Gridworld* environment consists of a $10 \times 10$ grid in which 2 objects of each of 10 different types are placed, so 20 objects in total. Each object type gives a reward uniformly sampled from $[-1, 1]$ when standing on it, while floor tiles give 0 reward. Between each two cells, with probability 0.3 there is a wall. The environment has a discrete states space with 100 states and a discrete action space with 5 actions: *north*, *east*, *south*, *west*, and *stay*. The dynamics are deterministic. The initial position of the agent is randomly selected but fixed for one instance of the environment.

For the GP model of the reward we choose a kernel

$$k(s, s') = \begin{cases} 1 & \text{if in } s \text{ and } s' \text{ the agent is standing on} \\ & \text{the same object type} \\ 0 & \text{else} \end{cases}$$

so that the model learns a reward for each of the object types independently.

The *Gridworld* environment shows that to select informative queries it is important to consider the reachable space in the environment. Queries of objects that are not reachable from the agent's initial position are not informative.

### D.3.4 Driver

We implement the *Driver* environment based on code provided by Sadigh et al. (2017) and Bıyık et al. (2020b). Here, we provide a brief description of the dynamics and features of the environment. For more details, refer to our implementation, or Sadigh et al. (2017).

The *Driver* environment uses point-mass dynamics with a continuous state and action space. The state $s = (x, y, \theta, v)$ consists of the agent's position $(x, y)$, its heading $\theta$, and its velocity $v$. The actions $a = (a_1, a_2)$ consist of a steering input and an acceleration. The environment dynamics are defined as

$$\begin{aligned} s_{t+1} &= (x_{t+1}, y_{t+1}, \theta_{t+1}, v_{t+1}) \\ &= (x_t + \Delta x, y_t + \Delta y, \theta_t + \Delta\theta, \text{clip}(v_t + \Delta v, -1, 1)) \\ (\Delta x, \Delta y, \Delta\theta, \Delta v) &= (v\cos\theta, v\sin\theta, va_1, a_2 - \alpha v) \end{aligned}$$

where $\alpha = 1$ is a friction parameter, and the velocity is clipped to $[-1, 1]$ at each timestep.

The environment contains a highway with three lanes. In addition to the agent, the environment contains a second car that changes from the right to the middle lane, moving on a predefined trajectory. The reward function is linear in a set of features

$$f(s) = (f_1(s), f_2(s), f_3(s), f_4(s), 1)$$

where $f_1(s) \propto \exp(d_1^2)$, $d_1$ is the distance to the closest lane center, $f_2(s) \propto (v-1)^2$, $f_3(s) \propto \sin(\theta)$, and $f_4(s) \propto \exp\left(-d_2^2 - cd_3^2\right)$ where $d_2$ and $d_3$ are the distance between the agent's car and the other car along the $x$ and $y$ directions respectively, and $c$ is a constant. Note that these features correspond to the version of the environment that Bıyık et al. (2020b) use and differ slightly from Sadigh et al. (2017). Reward functions for the environment are sampled from a Gaussian with zero mean and unit covariance. The first 4 features are normalized before the constant is appended.

The *Driver* environment uses a fixed time horizon $T = 50$, and policies are parameterized by 5 actions that are each applied for 10 time steps. For solving the environment we optimize over these policies using an L-BFGS-B solver as proposed by Sadigh et al. (2017). We additionally use the set of candidate policies $\Pi_c$ as a lookup table to reduce the variance of this solver. Whenever $\Pi_c$ contains a policy that is better for a given reward function than the one returned by the solver, we choose this one instead. This was first proposed by Wilde et al. (2020).

### D.3.5 MuJoCo Corridor

Our MuJoCo *Corridor* environments are based on code of the maze environments by Duan et al. (2016). Our "maze" is a corridor of 13 cells. The robot starts in the leftmost cell and one of the cells is a fixed goal cell. The true reward function, that is not directly available to the agent, rewards the agent proportional to its velocity in positive x-direction if the agent is before the goal, and rewards the agent proportionally to its velocity in negative x-direction if the agent is

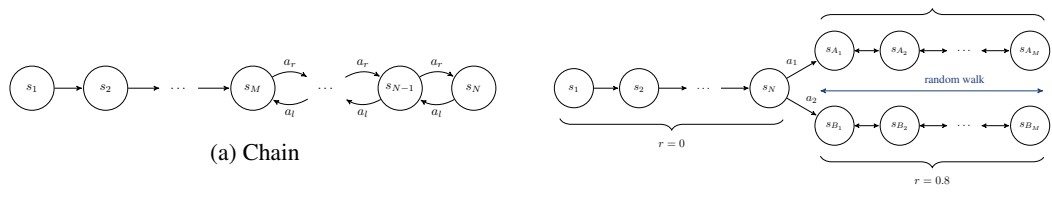

(a) Chain

(b) Junction

Figure 5: Illustration of the *Chain* and *Junction* MDPs. In the *Chain* MDP (a), the agent moves right with probability 1 from state $s_1$ to $s_M$, and the agent can move deterministically left and right from state $s_M$ to state $s_N$ with $N > M$. The reward function is sampled from the GP prior with a square-exponential kernel. We choose $M = 10$ and $N = 20$. In the *Junction* MDP (b), the agent moves right with probability 1 from state $s_1$ to $s_N$. In $s_N$, the agent can take the upper or lower path, denoted with $A$ and $B$, respectively. Along both paths, the agent moves either left or right with probability $0.5$. In the lower path, the reward of all states is $0.8$. In the upper path, the highest reward is greater than $0.8$, but the average reward is less than $0.8$. We choose $N = 15$ and $M = 5$. For both environments, the initial state distribution is uniform over the state space.

past the goal. This provides a reward function that is harder to learn than just moving in one direction. We encode this reward function as a linear function of a set of features

$$f(s) = (v_x I_1, v_y I_1, I_1, \ldots, v_x I_{13}, v_y I_{13}, I_{13})^T \in \mathbb{R}^{39}$$

where $I_k$ are indicator features that are $1$ if the agent is in cell $k$ and $0$ otherwise, and $v_x$ and $v_y$ are the x- and y-velocity of the center of mass of the *Swimmer*.

We use *augmented random search* (Mania et al., 2018) with linear policies to solve the environment for a given reward function. The policy is linear in a seperate set of features than the reward function. The features for the policy are based on the standard features provided by the *Swimmer* environment, extended by an indicator feature for the Swimmer being in each of the cells.

# E    Implementation details of `IDRL` with neural network reward models

This section provides more details on our implementation of `IDRL` using DNN reward models. Algorithm 5 shows pseudocode of the full algorithm.

## E.1    Training the reward model

We represent the reward function as a function of observations $o$ and actions $a$, using a DNN model which we can write as $\mu_{\hat{r}}(o, a) = \theta^T f_\phi(o, a)$. We conceptually separate the model into a feature representation $f_\phi(o, a)$ parameterized by weights $\phi$ and a linear function $\theta$. In practice $\theta$ is the last layer of the DNN and $\phi$ and $\theta$ are trained jointly.

We train the model on a dataset of pairwise comparisons of clips of trajectories. Let $\sigma = (s_1, a_1, \ldots, s_L, a_L)$ denote a sequence of state-action pairs of lengths $L$, and let $r(\sigma) = \sum_{i=1}^{L} r(s_i, a_i)$ be the sum of rewards over the sequence. We make queries $q_i = (\sigma_{i1}, \sigma_{i2})$, $y_i \in \{-1, 1\}$ that compare two such sequences of state-action pairs. Similar to Christiano et al. (2017), we choose the Bradley-Terry observations model for the comparisons:

$$p(\sigma_1 > \sigma_2 | \theta, \phi) = \frac{\exp(\hat{r}(\sigma_1))}{\exp(\hat{r}(\sigma_1)) + \exp(\hat{r}(\sigma_2))} = \frac{1}{1 + \exp(-(\hat{r}(\sigma_1) - \hat{r}(\sigma_2)))}$$

which is equivalent to Logistic regression on $\hat{r}(\sigma_1) - \hat{r}(\sigma_2)$. We use gradient descent to minimize the negative log-likelihood of the data under this observation model

$$\hat{\phi}, \hat{\theta} \in \operatorname{argmin} \mathcal{L}(\theta, \phi)$$

$$\mathcal{L}(\theta, \phi) = -\log p(\mathcal{D}|\phi, \theta) + \lambda(||\theta||^2 + ||\phi||^2)$$

where we use $\ell_2$-regularization which corresponds to a Gaussian prior on the weights. Because the Bradley-Terry model is invariant to shifting the reward function, we use DNN layers without biases. Additionally, we normalize the output of the model when using it to train policies.

**Algorithm 5** *Information Directed Reward Learning* (IDRL) using DNNs and SAC to train policies. The algorithm maintains a currently optimal policy $\bar{\pi}^*$ as well as a set of candidate policies $\Pi_c$. The reward function is represented with a neural network feature function parameterized by $\phi$, and a Bayesian linear model given by mean $\theta$ and covariance $H^{-1}$. The main training loop alternates between updating $\bar{\pi}^*$, querying new samples and updating the candidate policies. To select queries, the algorithm implements the IDRL objective, where line 21 corresponds to equation (1) and line 27 corresponds to equation (2). Line 32 updates the feature representation and estimates the MAP of the posterior, and line 33 approximates the posterior using a Laplace approximation. This pseudocode omits all hyperparameters that control how often new samples are queried, how many samples are queries, and how often and for how many steps all models are updated.

---

1: Initialize policy $\bar{\pi}^*$
2: Initialize candidate policies $\Pi_c \leftarrow \{\pi_1^c, \dots, \pi_n^c\}$
3: $\mathcal{D} \leftarrow$ initial dataset
4:
5: Initialize DNN reward model $\mu_{\hat{r}}(o, a) = \theta^T f_\phi(o, a)$ with $\phi$ random weights, and $\theta = 0$
6: Update DNN parameters, $\phi$ and $\theta$, via supervised learning on $\mathcal{D}$
7: $H \leftarrow \left[\nabla^2 \log\ p(\theta|\mathcal{D}, \phi)\right]\Big|_\theta$
8:
9: **while** not done **do**
10:     Train policy $\bar{\pi}^*$ on reward $r_i(o, a) = \theta^T f_\phi(o, a)$ using SAC
11:
12:     **if** update candidate policies **then**
13:         Sample $\boldsymbol{\omega}_1, \dots, \boldsymbol{\omega}_n$ from $\mathcal{N}(\theta, H^{-1})$
14:         **for** $i \in \{1, \dots, n\}$ **do**
15:             Train policy $\pi_i^c$ on reward $r_i(o, a) = \boldsymbol{\omega}_i^T f_\phi(o, a)$ using SAC
16:             Estimate state visitation frequency $\mathbf{f}^{\pi_i^c}$ using Monte-Carlo rollouts
17:         **end for**
18:     **end if**
19:
20:     **if** query samples **then**
21:         $\pi_1, \pi_2 \in \mathrm{argmax}_{\pi, \pi' \in \Pi_c} (\mathbf{f}^\pi - \mathbf{f}^{\pi'})^T H^{-1} (\mathbf{f}^\pi - \mathbf{f}^{\pi'})$
22:         $\mathbf{v}_{\pi_1, \pi_2} \leftarrow \mathbf{f}^{\pi_1} - \mathbf{f}^{\pi_2}$
23:         Roll out policy $\bar{\pi}^*$ and collect a set of candidate queries $\mathcal{Q}_c$
24:         **for** $q_i \in \mathcal{Q}_c$ **do**
25:             Compute Hessian with the expected response to $q_i$:
26:             $H_{q_i} \leftarrow \left[\nabla^2 \log\ p(\theta|\mathcal{D} \cup \{(q_i, \hat{y}_i)\}, \phi)\right]\Big|_\theta$
27:             $u(q_i, \mathcal{D}) \leftarrow -\mathbf{v}_{\pi_1, \pi_2}^T H_{q_i}^{-1} \mathbf{v}_{\pi_1, \pi_2}$
28:         **end for**
29:         Sort queries by $u(q_i, \mathcal{D})$ and select $k$ queries $\{q_1, \dots, q_k\}$ with the largest values
30:         Make queries $\{q_1, \dots, q_k\}$ and observe $\{y_1, \dots, y_k\}$
31:         $\mathcal{D} \leftarrow \mathcal{D} \cup \{((q_1, y_1), \dots, (q_k, y_k))\}$
32:         Update DNN parameters, $\phi$ and $\theta$, via supervised learning on $\mathcal{D}$
33:         $H \leftarrow \left[\nabla^2 \log\ p(\theta|\mathcal{D}, \phi)\right]\Big|_\theta$
34:     **end if**
35: **end while**
36: **return** $\bar{\pi}^*$

---

To compute the IDRL objective, we need a Bayesian posterior. We fix the features $f_{\hat{\phi}}$, and perform Bayesian regression to approximate the posterior $p(\theta|\mathcal{D}, \hat{\phi})$. To this end, we consider $\hat{\theta}$ to be the mode of this posterior, and compute a Laplace approximation:

$$p(\theta|\mathcal{D}, \hat{\phi}) \approx \mathcal{N}(\hat{\theta}, H^{-1})$$

$$H = \left[\nabla^2 \log p(\theta|\mathcal{D}, \hat{\phi})\right]\Big|_{\theta=\hat{\theta}}$$

where $H$ is the Hessian of the log-likelihood at point $\hat{\theta}$. The Laplace approximation is a very basic technique for approximate inference; however, it is convenient in our case because it approximates the posterior as a Gaussian distribution. This means we can compute the entropy and information gain of this distribution similarly easy as for a GP model.

## E.2 Hyperparameter choices

**Neural network model.** We use the same network architecture as Christiano et al. (2017): a two-layer neural network with 64 hidden units each and leaky-ReLU activation functions ($\alpha = 0.01$). For training we use $\ell_2$-regularization with $\lambda = 0.5$.

**Policy training.** We use the `stable-baselines3` implementation of SAC (Raffin et al., 2019), with default hyperparameters. For training the policy, we append a feature to the observations that measure the remaining time within an episode, $f_t = (t_{\max} - t)/t_{\max}$, where $t$ is the current time step in an episode and $t_{\max}$ is the episode length. Adding this feature tends to speed up training significantly in the MuJoCo environments. We do not add this feature for learning the reward function. Policies are trained for $10^7$ timesteps in total.

**Sampling rate.** We provide 25% of samples to the reward model before starting to train the policy, and during training provide samples at a sampling rate proportional to $1/T$. Concretely, if $N_s$ samples are provided in $N_b$ batches over the course of training, the $i$-th batch will contain $\frac{N_s}{H_{N_b}} \cdot \frac{1}{T}$ samples, where $H_n$ is the $n$-th harmonic number.

**Candidate policies.** We maintain a set of 3 candidate policies, that are each updated $10^7$ timesteps, as the main policy. The candidate policies are updated in regular intervals, which are controled by a hyperparameter $N_p$. Over the course of training, the candidate policies will be updated $N_p$ times using $10^7/N_p$ timesteps each time.

**Hyperparameter tuning.** We only tuned two hyperparameters explicitly: $N_b$, the number of batches of training samples the model gets during training, and $N_p$ the number of times the candidate policies are updated during training. We selected all other hyperparameters after preliminary experiments and to be as similar as possible to Christiano et al. (2017). We first tuned $N_b$ using a random acquisition function and values in $\{10, 100, 1000, 10000\}$, and chose $N_b = 1000$ which gave the best performance evaluated over 5 random seeds. We choose the same $N_b$ for all acquisition functions. Then, we tuned $N_p$ for the IDRL acquisition function and values in $\{10, 100, 200, 400, 600, 800, 1000\}$. We chose $N_p = 100$ which lead to best performance evaluated over 5 random seeds. All hyperparameters were only tuned on the HalfCheetah environment.

## E.3 Comparison of our Deep RL setup to Christiano et al.

In this section we point out differences in our Deep RL setup compared to Christiano et al. (2017). Some of the modifications are necessary for applying IDRL. Other differences result from us not being able to reproduce the exact environments and hyperparameters because Christiano et al. do not provide code of their experiments.

**Reward model.** Christiano et al. model the reward function with an ensemble of DNNs. We learn a feature representation using a single DNN, and combine this with a Bayesian linear model which makes computing the IDRL objective more straightforward.

**Policy learning.** We use SAC while Christiano et al. use TRPO for learning the policy. We chose SAC because it is significantly more sample efficient in MuJoCo environments.

**Sampling rate.** Christiano et al. provide 25% of total samples to the model intially, and provide the rest of the samples at an adaptive sampling rate which they choose to be "roughly proportional to $2 \cdot 10^6/(T + 2 * 10^6)$" (Christiano et al., 2017, App. A.1), where $T$ is the number of environment interactions so far. Unfortunately, they do not provide enough information to exactly reproduce their sampling schedule. Instead, we simplify the schedule to be proportional to $1/T$.

**Clip length.** Christiano et al. query comparisons between clips that "last 1.5 seconds, which varies from 15 to 60 timesteps depending on the task"(Christiano et al., 2017, App. A.1). Unfortunately, they do not specify the framerate used for each task, so we can not reproduce the exact clip lengths. Instead, we simply choose a length of 40 timesteps for each environment which is roughly in the middle of the range they provide.

**Observations.** From their paper it is unclear whether Christiano et al. (2017) include the agents position in the observation in locomotion environments such as the HalfCheetah. Note, that including the observation makes the reward learning task much easier because the reward function is linear in the change of the agent's x-position. Therefore, we do not include the position in the observation which we use to predict the reward function.

**Penalties for termination.** Most of the standard MuJoCo environments have termination conditions, that, e.g., terminate an episode when the robot falls over. Such termination can leak information about the reward, i.e., longer episodes are better. Therefore, Christiano et al. replace "these termination conditions by a penalty which encourages the parameters to remain in the range" (Christiano et al., 2017, App. A). Unfortunately, they do not specify the exact penalties they use. We also remove the termination condition, but replace it with a bonus for "being alive" which is implemented in the version 3 environments of OpenAI Gym.

# F  Additional experimental results

## F.1  Experiments in small environments

In Figure 6, we provide more detailed results of our experiments in small environments. This includes experiments in the *Gridworld* environment, but also the *Chain* and *Junction* environments that were not presented in the main paper. The results confirm the conclusion we presented in the main paper: by focusing on queries that are informative about which policy is optimal, IDRL is able to learn better policies with fewer queries than the baselines.

## F.2  Ant Corridor

Figure 7 shows results in the *Ant-Corridor*, using the same experimental setup as the results in Figure 3b in the main paper, which shows results in the *Swimmer-Corridor*.

## F.3  Deep RL experiments in individual environments

In Figure 8 we provide learning curves for the individual MuJoCo environments that were aggregated to create Figure 4 in the main paper. To aggregate the results we normalized the return in each environment:

$$G_{\mathrm{norm}}(\pi) = 100 \cdot \frac{G(\pi) - G(\pi_{\mathrm{rand}})}{G(\pi^*) - G(\pi_{\mathrm{rand}})},$$

where $\pi_{\mathrm{rand}}$ is a policy that samples action uniformly at random, and $\pi^*$ is an expert policy trained using SAC on the true reward. This results in a score that is $0$ for a policy that performs as well as a random policy, and $100$ for a policy that matches an expert performance. In Figure 4 this score is averaged over all environments.

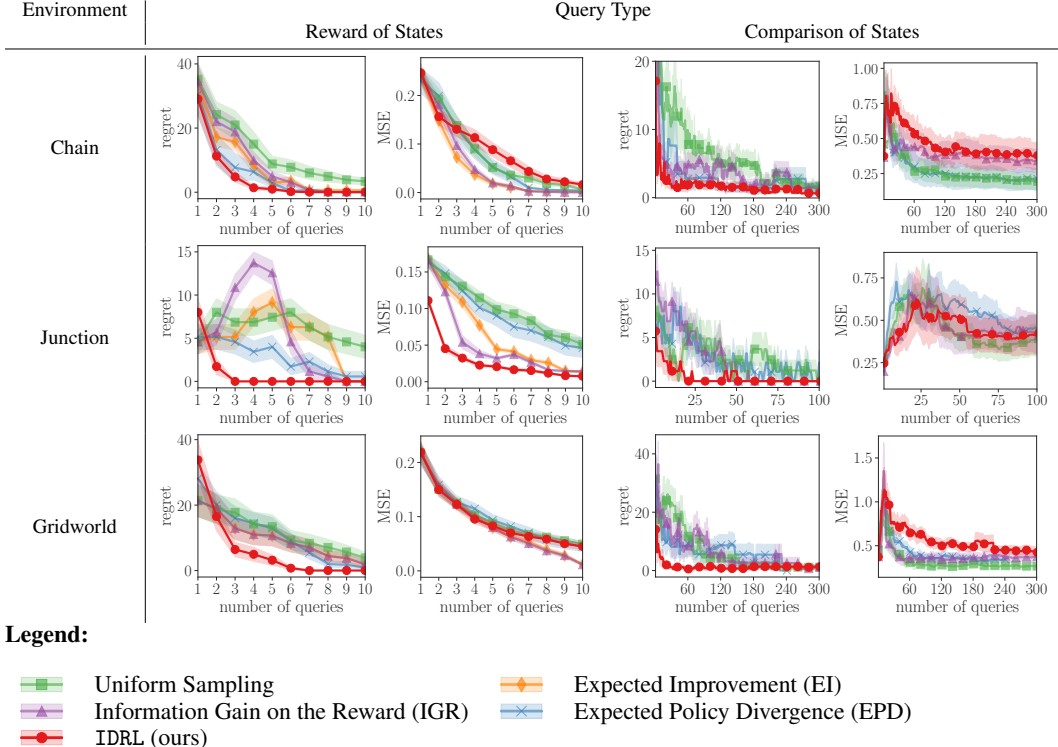

Figure 6: Learning curves for the *Chain*, *Junction* and *Gridworld* environments for two query types: the reward of individual states and comparisons of states. For each setting, one plot shows the regret of a policy trained using the reward model and a second plot shows the mean squared error (MSE) of the reward model over the whole state space. The plots show mean and standard error across 30 random seeds. Across all environments, IDRL learns a better policy with fewer queries. However, the MSE measured on the entire state space is usually worse, because IDRL focuses on regions of the state space that are informative about the optimal policy.

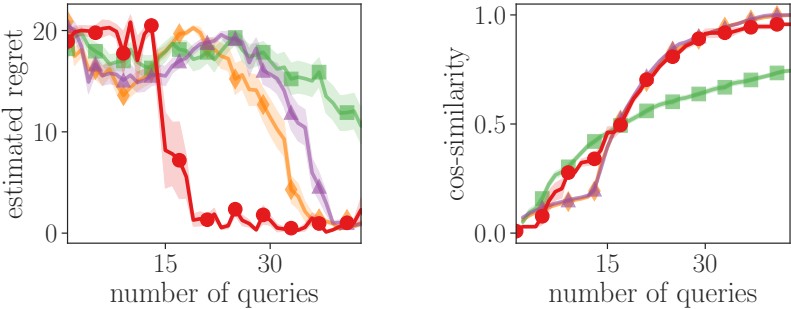

Figure 7: Regret and cosine similarity in the *Ant-Corridor* environment, comparing IDRL (⬤) to IGR (▲), EI (◆), and uniform sampling (■). The experimental setup is exactly the same to the results shown in Figure 3b in the main paper. The *Ant-Corridor* environment is the same as the *Swimmer-Corridor*, only with a different robot.

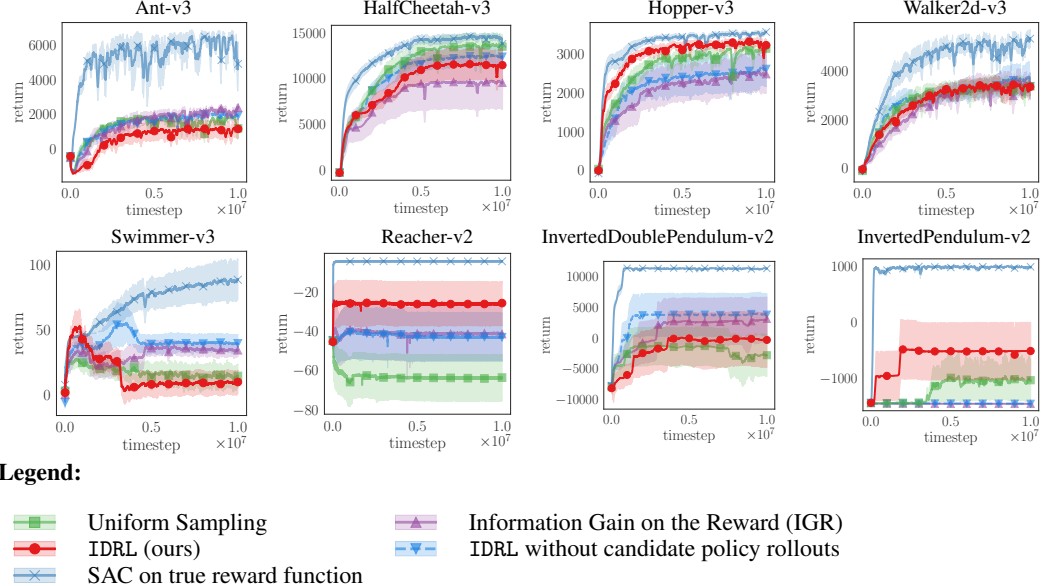

Figure 8: Return of policies trained using a model trained from 1400 comparison for each of the MuJoCo environments. Figure 4 in the main paper shows the normalized average over all environments.