# OpenReview forum: "Information Directed Reward Learning for Reinforcement Learning"
_NeurIPS.cc/2021/Conference — NeurIPS 2021 Poster_

### Official Review · Reviewer_nVyN · 2021-07-08

**Rating:** 7
**Confidence:** 4

**Summary:**

Information Directed Reward Learning (IDRL) actively selects queries that are most informative at distinguishing policies in terms of their value. By contrast, prior work has typically tried to maximize information gained about the reward function itself -- but some parts of the state space may be unreachable, or never visited by any plausible optimal policy, and so learning about them is irrelevant for task performance. IDRL has a pleasing theoretical motivation, although practical implementations of it require considerable use of approximations and heuristics. Additionally, IDRL is compatible with a broad variety of reward learning queries (e.g. preference comparisons, labeling trajectories with returns) whereas most prior work focuses on a single feedback modality. Experimental results show significant improvements in sample efficiency from use of IDRL relative to a variety of baselines.

**Limitations And Societal Impact:**

One significant limitation of the work that is unfortunately not discussed is that it may harm generalization relative to other active learning criteria, such as maximizing information gain over the reward. You highlight generalization as a significant advantage of reward learning (line 44 onwards): "policies tend to generalize poorly between environments" while "reward functions are a more robust representation of desired behavior". But if you seek only information that lets you discriminate between policy return, you are liable to lose much of this generalization benefit. Indeed, consider Figure 1: IDRL specifically avoids querying about the pear. If at deployment the wall around the pear is removed, an IDRL agent will be highly uncertain what to do, whereas random sampling or better yet EVR would have learned about the pear.

There's a fundamental tradeoff here, and I think IDRL may well be the right choice for many applications, but it seems disingenuous to simultaneously claim the benefits of reward learning in terms of generalization and systematically avoid gathering information that would help generalize.  I did find it interesting that the cosine similarity (Figure 3) nonetheless was higher or comparable to existing method, but I would need to see comparison of learned to ground-truth rewards in more environments to be convinced, or explicit test of transfer to new environments.

**Main Review:**

The paper tackles an important problem, sample-efficient reward learning, with a novel approach that has some theoretical justification and promising empirical results. The presentation is in general very clear, and the experiments seem to be carefully executed: e.g. I was pleased to see the environment termination conditions were removed in section 7.5, which is a common pitfall for work in this area.

My main hesitation lies in the scalability of the approach. The GP approach, while having pleasing theoretical properties, seems intractable to scale without significant hand-designed feature engineering -- e.g. section 7.4 required pre-training fixed policies for Ant, and there was a relatively simple mapping from observations to reward. The deep RL approach seems potentially more scalable, but the approximations used are so substantial that the method loses much of its theoretical justification. The empirical performance of deep IDRL goes some way to allaying this concern, but the improvements are modest and inconsistent between environments, on occasion even doing worse than random sampling.

Overall I vote to accept the paper. In particular, the key idea of choosing queries based on how this affects the return achieved seems like an insight that will prove useful in the long-term. I am unsure if the IDRL method itself will see adoption, but it is certainly plausible, especially as Bayesian NNs improve the deep IDRL method may be able to scale.

# Originality

Moderately original. The algorithm is new, but operates in the existing (but understudied) paradigm of active reward learning. The key idea of focusing on information gain on policy-relevant factors rather than reward itself has been considered in other contexts before (e.g. Russo and Van Roy, 2014 which is cited in the appendix). But as far as I know it has not been applied to reward learning in general sequential decision making problems before, and the focus on difference in return between policies is new (although I'm not convinced it's quite the right objective).

# Quality

The paper generally seemed to be carefully executed. I did not check the proofs in the appendix but all claims and derivations seemed plausible. Details of the experimental setup were for the most part clear, the baselines were reasonable and when a direct comparison could not be made the authors highlighted this limitation.

# Clarity

While the paper was in general easy to read, there were a few parts that seem confusing or ambiguous:
  - Use of the phrase "potential reward function" at line 248 and elsewhere is ambiguous. I think you mean something like a candidate reward function. But it could also be misunderstood to mean a reward function based on potential shaping. I would suggest rephrasing.
  - It is unclear what the reward function in section 7.4 actually is -- can you add more detail or least reference a relevant section in the appendix? In particular, I am not sure what is meant by a "one dimensional grid". Furthermore, you say it is "different across the grid" -- different in what way?
  - You say IDRL supports multiple query types, but it's unclear to me exactly how flexible it is. Could you choose between queries of different types (e.g. preference comparisons vs reward labels) on the basis of informativeness? If so, why does no experiment do this? Multi-modal feedback seems particularly promising to me, as some feedback sources (e.g. demonstrations) are likely to be more sample-efficient at the start of training when uncertainty everywhere is high, whereas other feedback sources (e.g. preference comparison) are likely to be more informative towards the end of training when it is necessary to disambiguate specific areas. This could be a key advantage of IDRL that I don't see any mention of currently.

There were also a few more minor points, which I discuss at the end of the review.

# Significance

The problem domain is important. I find it unlikely that IDRL will be of direct use to practitioners today due to the computational cost and complexity of implementation -- it may often be cheaper to just gather more data -- but I expect this to improve over time.

-----
# Minor typos etc

- Figure 1: overall I really liked this figure. However I initially didn't parse the square around the pear as being an additional wall, and so was confused why there were only two plausible optimal policies. It didn't take long to figure out, but it might be worth being explicit in the caption: e.g. “It can only move 4 timesteps in the gridworld, *cannot move through walls (in black)* and collecting more food is always better.”

- Line 72: “(MDPs, Puterman, 2014)” is confusing – is MDP an author? Use a semicolon instead to separate: (MDPs; Puterman, 2014)”.

- Line 79: $r_t$ is not defined. Although clear from context, this is still worth defining, e.g. you could add it in the sentence starting “In an MDP, ...”

- Line 114: what is $\hat{p}^*$? I think this is meant to be $\hat{\pi}^*$?

- References: some capitalization issues, likely due to missing {} around words to preserve capitalization in your .bib file. Line 396: “Openai gym”, line 423: “atari”. Please proofread this section.

**Time Spent Reviewing:**

2.5

---

> ### Author Response · Authors · 2021-08-09
> **Addressing scalability and generalization**
>
> Thanks for the detailed and thoughtful review! Below, discuss the two main concerns you mentioned regarding our work, namely scalability and generalization.
>
> You are right that the full GP-based implementation of IDRL without hand-designed features would be intractable for many practical applications, and that our Deep RL implementation of IDRL performs better in some environments than others. However, there is a wide spectrum of alternative implementations of IDRL using some, but not all the approximations. For example, to avoid hand-engineered features, one could just use the GP-based model together with learned basis functions, but not embed the reward learning process into a policy optimization loop. This would give a different trade-off between theoretical justification and scalability, and likely lead to better performance than the full Deep RL approach. Unfortunately, we could not investigate all possible such variants of IDRL in our experiments, and leave this to be investigated in future work.
>
> We agree that the generalization of the learned reward functions can be important and that  this aspect is not sufficiently discussed in the paper. There is indeed a fundamental tradeoff between learning the reward function to find a good policy in a specific environment, and learning a reward function to generalize to other environments. If we don’t have information about the deployment environment, active learning methods that query uniformly across the state space might very well be the best choice. However, if prior knowledge about the distribution of possible environments is available, it is clearly possible to do better. IDRL, as described in the paper, is not concerned with generalization between environments, but aims to learn a model of the reward function that can be used to identify an optimal policy as quickly as possible for a single environment. IDRL could likely be extended to work with a prior distribution over environment dynamics as well, but we have not addressed this in this paper. We will mention this point in our discussion of limitations and potential future work.
>
>
> Regarding the other confusions you pointed out:
>
> The term “candidate reward function” is much better and less confusing than “potential reward function”. We will change this terminology, thanks for the suggestion.
>
> We will improve the description of the reward function in Section 7.4. In particular, it seems like the “grid” terminology was confusing, and we will update that. You can find detailed information about the reward function in Appendix D.3.5. There, we explain that when we refer to the “one-dimensional grid” we mean that we split the x axis along which the agent has to move into equally spaced regions (“grid cells”) with a different reward function in each of them. In each region the agent is rewarded for moving in a specific direction, but the direction changes depending on which region the agent is in (left of, right of or at the goal location).
>
> Regarding multiple query types: IDRL can be used for different query types as long as a Bayesian model can be updated with the queries and the necessary information gain can be computed or approximated. It is indeed a big advantage of a method based on information gain, that the values of the acquisition function are comparable between different query types. However to select between queries of different types, one would have to take the different cost of making these queries into account. For example, a demonstration will usually contain much more information than a comparison query, but it will also be more costly to provide. We do not consider query post in this work, which is why we focus on experiments with a single query type.
>
> Will will address all additional typos, and minor issues in the revised paper. Thanks for making us aware of these.

---

> > ### Comment · Reviewer_nVyN · 2021-08-12
> > **Reply**
> >
> > Thanks for the response. I agree there are a broad range of alternative implementations of IDRL varying in their level of approximation and tractability, and I am excited to see future work that might improve scalability.
> >
> > I look forward to seeing a revision of the paper with generalization discussed as a limitation. Having a prior distribution over environment dynamics would be interesting, although I agree is likely out of scope for this paper.
> >
> > It's great that IDRL can work with multiple query types.

---

### Official Review · Reviewer_kbe1 · 2021-07-15

**Rating:** 8
**Confidence:** 3

**Summary:**

The authors introduce a query selection strategy for reward learning in reinforcement learning. This strategy can use different query types and is not restricted to pairwise trajectory queries, as many other approaches. Additionally, the strategy is also considering the environment dynamics. The papers main contribution is a method, that reduces the number of required queries, besides the aforementioned aspects. The authors also show, that the approach is also applicable to Deep RL techniques.

**Limitations And Societal Impact:**

The authors addressed several limitations, but there is quite a bit room for improvement. This mostly relates to section 7.5 (as stated in the main review). It is especially interesting if the mentioned problems are induced by the DNN approach,  the limited candidate update rate or the higher dimensionality. Multiple ideas come to mind: Using scalable, exact GP approaches [Exact Gaussian Processes on a Million Data Points, Wang, 2019], Kernel Neural Networks with GPs [Simple and Principled Uncertainty Estimation with Deterministic Deep Learning via Distance Awareness, Liu, 2020] or running SAC from scratch often. However, i acknowledge, that this would introduce substantial runtime demands, making it an interesting study for future work.
Lastly, evaluations concerning the non-linear query capabilities should be added

**Main Review:**

The algorithm is based on well known entropy-based learning approaches, but combines this idea with policy awareness. This is an interesting idea and a relevant step forward, despite the fact that the elements themselves are not novel. As this algorithms builds on principles with well established theory, it can be considered sound and is clearly described. Related work is sufficiently covered, differences mentioned and the experimental section is also mostly well structured and understandable.

The novel sections (wrt. "Improvements Made") are a clear improvement of the paper, but could be improved. As example, the algorithm 5 from the appendix is relevant for understanding the DNN setup and should be moved to the main paper, or at least, explained in more detail. A more substantial limitation is the discussion of the results from the Deep RL setting (7.5). From the appendix, it is clearly visible that IDRL is subject to failure modes (e.g. Swimmer-v3) or premature convergence (Ant-v3, InvertedDoublePendulum-v2). The same is true for most other approaches, but potential reasons are not really discussed. The section (together with 7.4) is sufficient for fulfilling the claim, that it is computationally possible to scale the approach to higher dimensional problems, but performance remains in question.

The rest of the evaluation was performed on simpler domains, enabling the authors to disregard the scalability and exploration issues of the RL approach (by fully recomputing the candidate set multiple times). This is very interesting, as it allows to specifically determine the effect of the query selection strategy, which is the papers main contribution. It should also be positively considered, that they compare against a broad range of baseline methods. However, the capability of dealing with non-linear reward queries was not evaluated, but it is clear, that it is theoretically possible.



**Time Spent Reviewing:**

3

---

> ### Author Response · Authors · 2021-08-09
> **We will improve the discussion of the Deep RL implementation, results, and limitations**
>
> Thanks for the review, and in particular for taking the time to consider our empirical evaluation in great detail.
>
> We agree that the experiments using the Deep RL implementation of IDRL are somewhat limited, and that the results are not conclusive in all environments. As you said, these experiments aim to demonstrate that IDRL can be scaled up. It is likely that the performance could be improved further on some of the individual environments; however, we wanted to report average performance without too much parameter-tuning. We will expand the discussion of the results on individual environments in the main paper and discuss potential reasons for failure in some environments. Also, we will add a more explicit discussion of the limitations of the current Deep RL algorithms together with potential variations to be explored in future work.
>
> We will also improve the discussion of the implementation of our Deep RL-based variant of IDRL and provide more details in the main paper, although we might not be able to move Algorithm 5 to the paper due to space constraints.
>
> We are glad that you appreciated the rest of our experiments. Note that we did consider non-linear reward functions in some toy environments. However, these results only appear in Appendix F and are not discussed in the main paper. Depending on space constraints, we will either include these results in the main paper, or, at least, update the main text to explicitly mention them with a reference to the appendix.

---

### Official Review · Reviewer_2bor · 2021-07-15

**Rating:** 9
**Confidence:** 2

**Summary:**

The authors propose a Bayesian method for learning the reward function in an RL setting.  They study the setting in which the algorithm can query experts.  The form these queries can take is quite general.  They analyse their algorithm theoretically and empirically.  Their key insight is that queries should be focused on enabling the reward model to be used to find the best policy possible (as efficiently as possible), rather than decreasing the reward model’s uncertainty in general.


**Limitations And Societal Impact:**

Yes.

**Main Review:**

--BEGIN UPDATE--

After reading the other reviews and the author responses, I maintain my score; this is an outstanding paper.  I think the improvements the authors have promised will make it even stronger.

--END UPDATE--


The amount of details included about the experimental setup is outstanding (main paper and Sections D-F) and the additional experiments/theory in the appendix are very nice.  The experiments have outstanding breadth and depth and the results in Section 7.3 in particular were interesting and well-thought-out.

Small issues/edits:

-\mathcal D is first used about 5 lines before it is defined, it would be easier to read if the definition was moved before the first use.

-There are some definitions that can only be inferred or are stated informally, and should be more formally defined.  For example, the belief about the optimal policy, y hat and the set within which it exists, G hat (see next issue below; I’m not sure which “G” “G hat” estimates, a very literal reading makes me think it estimates the first definition, but the context makes me think it estimates the second definition), etc.

-The definition of G(pi) is overloaded; this is confusing and should not be done.  (This maybe contributes to the confusion in the point above.)  Edit: On my second read-through, it occurs to me that the second “definition” of G(pi) (on line 128) is actually not a definition, and that that notation denotes the dot product.  This would explain a lot of my confusion above. Two suggestions if this is the case: 1) use \coloneqq for definitions (for example, the definition of G in Section 3) to clearly distinguish definitions from equalities, and 2) tell the reader on line 128 that that notation denotes the dot product (at first, I assumed it was a vector or a tuple or a contention of vectors, or something like that; in other words I thought that G was being overloaded, and the new definition included f^pi and r rather than just their scalar dot product).

-Line 8 of algorithm 1 (I think it’s the update to the belief about the reward function) is a bit confusing to me at a high level; high-level intuition/clarification in the pseudocode or in the relevant text (the second to last paragraph of Section 4) would be nice.

-The error bars of Figure 2 are not specified (I assume they are standard error based on the other figures).

-Please specify the number of trials (random seeds, or “runs”) used for Figure 2.  (This would be a major issue, except that the bars, which I assume are standard error, see above, seem to indicate that there is a reasonable amount of statistical significance in this experiment.)

-The caption of Figure 6 in the appendix has a dead reference (to a section or a figure perhaps).

Major issue: Line 114: p^* is completely undefined. “I” (the mutual information?) is also undefined. Because of these undefined symbols, I’m not following the next parts (e.g., the “two undesirable properties”) as well as I would like.  Related minor suggestion: Assuming the latter is mutual information, perhaps there should be a little background on this; I suspect readers are more likely to know the basics of MDPs than the basics of information theory (and the authors have rightly included MDP basics).  This can be a short appendix section if there’s no room in the main paper.

Question/suggestion: Throughout the paper, you assume unlimited and cheap access to the environment and its transition function (minus the reward function), correct?  There’s nothing with this problem setting, but it might be worth explicitly mentioning this point in Section 1 or 3.

In summary, I strongly recommend an accept.  This recommendation is contingent on the authors addressing the (easily-fixed) issues I am concerned about.  However, my confidence in my assessment is low; so my assessment may change depending on the feedback of/discussion with the other reviewers.


**Time Spent Reviewing:**

~6-7

---

> ### Author Response · Authors · 2021-08-09
> **A few clarifications**
>
> Thanks a lot for the positive review, we are glad that you appreciated our work. The detailed list of confusions you encountered together with concrete suggestions will help us a lot in improving the paper. We will carefully consider and address all points. Here, we address only the major confusions:
>
> We agree that our notation of $G(\pi)$ could be made clearer. The notation in line 128 indeed denotes a dot product and is just a different description of the expected return of a policy as defined before, rather than overloading the definition. Thanks for the concrete suggestions for reducing the confusion. We will make sure that it is clear which statements are definitions, and we will clarify the dot product notation in line 128.
>
> Figure 2 shows standard errors over 30 random seeds, we will add this information to the caption.
>
> Line 114 is a typo, it is supposed to be $\hat{\pi}^*$ instead of $\hat{p}^*$. $\hat{\pi}^*$ refers to the belief about the optimal policy, as defined a few sentences before. Thanks a lot for pointing out this typo that crucially hinders understanding this part. The symbol $I$ refers to the information gain, which we agree could be stated more clearly here. We will make this more explicit, and also add a short section defining the information gain and giving a brief intuition.
>
> Regarding your questions about cheap access to the environment: yes, we assume cheap access to the environment or a simulator, and instead focus on improving the sample efficiency in terms of queries about the reward. It is a good suggestion to make this more explicit in the paper. When revising the paper, we will explicitly mention this in the problem setup, and also refer to it as a limitation in the conclusion section.

---

### Official Review · Reviewer_vsGF · 2021-07-16

**Rating:** 6
**Confidence:** 3

**Summary:**

This paper proposes a new active reward learning method called IDRL, designed to select the most informative query for identifying an optimal policy among a set of plausibly optimal policies. First, IDRL selects two candidate policies that maximize the entropy of the difference in expected returns given past queries. Then IDRL chooses queries that reduce the uncertainty of the return difference most among a set of candidate queries. Updating the reward model using the selected queries rather than queries maximizing the information gain about the reward stimulates identifying the optimal policy. Experimental results show that IDRL requires fewer queries than the considered baselines and works with different query types.

**Limitations And Societal Impact:**

The authors have addressed the limitations and potential societal impact.

**Main Review:**

(1) Originality and significance: \
This work changes the main focus of active reward learning from achieving a low estimation error of the reward to directly improving the policy which the reward model induces. This approach can be a new direction for further research in the active reward learning area.

(2) Quality: \
This paper is supported by theoretical background on Gaussian distribution and GP. IDRL scales to Deep RL setting with approximation the posterior as Gaussian using Laplace approximation, making the algorithm tractable (including computation of entropy and information gain).

Here are some related questions. \
Q1. Can you show any statistical test for the results in Figures 3 and 4? \
Q2. What is the performance tendency on the number of candidate policies (although too many candidates cannot be used due to the computational cost)?

(3) Clarity: \
Here are some questions for clarification. \
Q3. In Section 7.5, "it(IDRL) generates the candidate queries rolling out the currently optimal policy and the candidate policies(356-357)." On the other hand, Algorithm 5 of Appendix E.1 uses the currently optimal policy only (line 23)." Does the author say that the former is IDRL and the latter  'IDRL without candidate policy rollouts'?\
Q4. In Algorithm 5 of Appendix E.1, does line 25 mean that the Hessian is calculated after (virtually) updating DNN with $q_i$ and its predicted response $y_i$? \
Q5. In Section 7, "the agent's queries are answered with simulated feedback based on an underlying true reward function(219)." Could you clarify how 'simulated feedback(expert)' runs (or is implemented) in detail?


---------------- Post Rebuttal --------------------

I read the authors' response carefully and appreciate the detailed explanations. I am inclined to accept this paper.




**Time Spent Reviewing:**

12

---

> ### Author Response · Authors · 2021-08-09
> **Addressing Q1-Q5**
>
> Thanks for the review. We are glad you appreciated the originality of the problem and our approach. Let us directly address the open questions:
>
> Q1: In Figure 3 and 4 we plot the mean and standard error computed over 30 and 5 independent experiments with different random seeds, respectively. Assuming a normal distribution, the standard errors can be interpreted as confidence intervals for the population mean. Note that we follow the standard practice in Deep RL papers to report the learning curves over multiple experiments (e.g., [1,2]).
>
> Q2: We performed preliminary experiments indicating that the marginal benefit of considering additional policies decreases rapidly as a function of the number of candidate policies. In the experiments that we report in the paper, we mostly use 5 candidate policies, which we found to be a good trade-off between performance and runtime.
>
> Q3: You are correct that Algorithm 5 is missing the rollouts of the candidate policies. So in the current version Algorithm 4 is what we describe as “IDRL without candidate policy rollouts” in the main paper. However, we will update Algorithm 5 to include the candidate policy rollouts. Thanks for noticing this inconsistency.
>
> Q4: Yes the Hessian is computed for each “virtual” update using the predicted values. We will clarify this by adding a comment to line 25 in Algorithm 5.
>
> Q5: The details of the simulated feedback differ among the experiments. Numerical queries about the reward of single states or the return of (parts of) trajectories are simulated as observations of the “true” values perturbed by additive Gaussian noise. The responses to comparison queries in the highway experiment are noisy and simulated according to the linear observation model described in Appendix B. In the MuJoCo environment, the responses to comparison queries are noise-free, i.e., the same query will always result in the same observation of which trajectory is better according to the “true” reward . We do this to be consistent with [2]. These details are explained in the appendix; however, we agree that this should be made clearer in the main paper. We will add more details on the simulated feedback to Section 7 in the revised paper to address this.
>
> [1] Haarnoja, Tuomas, et al. "Soft actor-critic algorithms and applications." arXiv preprint arXiv:1812.05905 (2018).
>
> [2] Christiano, Paul, et al. "Deep reinforcement learning from human preferences." arXiv preprint arXiv:1706.03741 (2017).

---

### Decision · Program_Chairs · 2021-09-27

**Decision:**

Accept (Poster)

**Comment:**

This paper proposes a new active reward learning algorithm, where where the agent is required to reason about unknown reward function by querying an expert. The main idea is to select queries that maximize the information gain about the difference in return between policies with high uncertainties of their return difference. The results show that the proposed method can achieve similar or better performances with fewer queries and can work with different types of queries unlike the prior work.

All of the reviewers appreciated that the proposed objective that focuses on improving policy as opposed to reducing reward model error, which has been done in the prior approaches, is novel and technical sound. The paper also presents both a theoretically exact implementation and a scalable approximation for Deep RL. The results look solid in that the proposed method is much more query-efficient and can work with more types of queries compared to the baseline approaches. Overall, this is a solid and neat paper. Therefore, I recommend to accept the paper.